# Is drought protection possible without compromising flood protection? Estimating the potential dual-use benefit of small flood reservoirs in Southern Germany

Sarah Quynh-Giang Ho[1,2], Uwe Ehret[1,2]

[1]Institute for Water and the Environment - Hydrology, Karlsruhe Institute of Technology, Karlsruhe, 76133, Germany
[2]Center for Disaster Management and Risk Reduction Technology (CEDIM), Karlsruhe Institute of Technology, Karlsruhe, 76133, Germany

*Correspondence to*: Sarah Quynh-Giang Ho (sarah.ho@kit.edu)

**Abstract.**

As climate change drives intensification and increased frequency of hydrological extremes, the need to balance drought resilience and flood protection becomes critical for proper water resources management. Recent extreme droughts in the last decade in Germany have caused significant damages to ecosystems and human society, prompting renewed interest in sustainable water resources management. At the same time, protection from floods such as the catastrophic 2021 event in the Ahr Valley remain heavy in the public conscience. In the state of Baden-Württemberg in southwestern Germany alone, over 600 small (< 1 million m$^3$) to medium-sized (1-10 million m$^3$) reservoirs are currently operated for flood protection. In this study, we investigate the potential of different reservoirs for a dual flood-drought protection scheme that introduces a retention flow to store excess water for release in drought conditions, with the assumption that locations with more water available for storage will be better able to mitigate downstream streamflow drought. 30 reservoirs in Baden-Württemberg are selected based on their size according to the German design standard DIN19700 (where small reservoirs have capacities of roughly 50,000-100,000 m$^3$, medium have 100,000-1,000,000 m$^3$, and large have more than 1,00,000 m$^3$), their purpose (flood-only or multipurpose), and their relative water availability (expressed as the number of times the reservoir can be filled by the difference between the mean inflow and mean low flow). These reservoirs, despite their DIN19700 sizing categories, remain small in the context of global reservoir studies. Daily target releases for drought protection are proposed based on the 70$^{th}$ percentile exceedance of modeled inflows from the calibrated hydrological model LARSIM. The retention flow is optimized to maximize penalty reduction in a scenario of perfect knowledge of flooding by using meteorological observations as artificial weather forecasts in LARSIM. The results of different retention flows are then evaluated based on their adherence to the target releases and flood protection performance. Reservoirs were required to maintain the same level of flood protection under these modified rules. The optimized results were varied: there are reservoirs that can release up to 80 times their capacity with limited benefit for streamflow drought prevention; others that can reduce streamflow drought conditions and water deficits by almost 95% over a 24-year simulation period; and others that have potential but are limited by either the capacity or by constraints

for flood protection. There seems to be a trade-off between relative water availability to the reservoir and ability to alleviate drought conditions. We find that relative water availability at the reservoir has a strong relation to the amount of water a reservoir can release for drought protection, but fails to summarily describe the reservoir's potential impact on drought conditions downstream.

## 1    Introduction

Reservoirs and their operation are a critical part of drought resilience infrastructure. The ability of reservoirs to enhance low flows and therefore reduce drought conditions has been demonstrated by many studies (Padiyedath Gopalan et al., 2020; Shih and Revelle, 1994, 1995; Huang and Chou, 2005; Karamouz and Araghinejad, 2008; You and Cai, 2008b, a; Chang et al., 2019). Research on optimal reservoir operation rules for drought have often focused on the concept of hedging rules. Hedging rules assume that by storing water and creating a small deficit of water now, we can use that water mitigate the consequences of a heavy deficit later (Shih and Revelle, 1994). While several types of hedging rules exist, Draper and Lund (2004) found that, for most cases, a two-point hedging rule (where hedging storage begins at one point and ends at another) is optimal. Hedging rules have been applied for not only drought hedging operations (Chang et al., 2019; You and Cai, 2008b), but also for environmental benefits (Adams et al., 2017) and flood operation (Hui et al., 2016). Further research has also demonstrated that flood hedging is similar to that of hedging for water supply (Zhao et al., 2014) . The combination of the two objectives— storing water for drought and maintaining retention capacity for flood retention—is difficult due to their inherently competing nature, but is more effective when the trigger rules are variable throughout the year (Chang et al., 1995; Balley, 1997). However, the majority of these studies focus on large drinking water reservoirs with capacities on the scale of 100 million to 1 billion $m^3$—whether such conclusions would hold for small reservoirs is uncertain.

Small reservoirs have often been named as a potential decentralized solution to water scarcity in semi-arid and arid regions (Wisser et al., 2010; Jurík et al., 2018; Casadei et al., 2019; Liebe et al., 2007). These are reservoirs typically defined as having a dam height of ≤15 m, a surface area of < 0.1 $km^2$, and / or a storage volume of up to 1-2 million $m^3$ (Jurík et al., 2018; Casadei et al., 2019). Because they are smaller, they are cheaper to construct and maintain, and can be implemented in otherwise remote locations (Qadir et al., 2007). They can also be much more easily adapted to local conditions and can be managed locally (Venot and Krishnan, 2011). While they have a plethora of benefits, such as flood retention, ecosystem protection, and recreation (Jurik et al., 2015; Ogilvie et al., 2019; Liebe et al., 2007), the most common usage is to capture rainwater for supplementing agriculture. Research in Thuringia, Germany, has suggested that recommissioning small reservoirs could maintain or even increase crop yields in an uncertain future (Heinzel et al., 2022). In a global-scale analysis of their potential impact, small reservoirs in certain regions were estimated to potentially increase green water flow—in other words, agricultural water—by up to 1,100 $km^3$ per year, with an estimated ~35% increase in cereal production (Wisser et al., 2010). As climate change impacts destabilize traditional water availability patterns, decentralized small-scale solutions such as small reservoirs may play a role in mitigating these effects.

However, small reservoirs are not without their challenges. Small reservoirs may release water of reduced quality due to eutrophication within the reservoir (Jurík et al., 2018) and may even worsen water shortages in the long term by unsustainably increasing demand (Di Baldassarre et al., 2018). According to one study, managers across Ethiopia, Ghana, Burkina Faso, and Zambia consider many (anywhere from 25-70%) of their small reservoirs to be performing poorly (Venot et al., 2012). For example, implementations in Ghana—while overall well-received by the local farmers for their plethora of benefits—were

found to have no statistically significant increase in the income of vegetable farmers (Acheampong et al., 2018). An analysis of 56 small reservoirs in Tunisia similarly showed that 16 of the reservoirs showed negligible benefits to the local agriculture (Ogilvie et al., 2019). Proposed reasons for the suboptimal operation of these reservoirs include insufficient inflow to the reservoir (Berhane et al., 2016); siltation, seepage, and evaporation losses (Acheampong et al., 2018; Mady et al., 2020); structural damage due to lack of maintenance (Berhane et al., 2016; Jurík et al., 2018; Casadei et al., 2019); and

mismanagement due to poor organizational capacity at the local management level (Venot et al., 2011; Acheampong et al., 2018). Despite these challenges, the potential additional water provided by small reservoirs is still extremely valuable for enhancing the resilience of local water resources in drought, especially in the context of rainwater harvesting via flood retention (Qadir et al., 2007); however, the potential benefits of these strategies in water-rich countries like Germany remains underresearched.

More than 800 reservoirs in the German southwestern state of Baden-Württemberg exist today, with total capacities ranging from as small as 200 m$^3$ to almost 43 million m$^3$. 90% of these reservoirs have dams less than 15 meters in height. The German reservoir design standard DIN 19700 (Lubw, 2007) categorizes these reservoirs by dam height and capacity into large, medium, small, and very small reservoirs (see Table 1). In the global context, the majority of these "small" and "medium" would be small reservoirs. Many of the "large" reservoirs in Baden-Württemberg are just above the cutoff and remain quite small in

comparison to typical large dams in the literature, which often have capacities that are at least an order of magnitude larger, generally 100 million to 1 billion m$^3$ (Consoli et al., 2007; Cañón et al., 2009; Liu et al., 2020). Henceforth we adopt the DIN 19700 size definitions as descriptors for reservoir sizes with the understanding that these refer to small and, at most, mid-size reservoirs on the global scale. Historically, flooding has been the major hydrological problem in the region: over 650 of the existing reservoirs are built for flood retention. Other uses include nature conservation, energy production, recreation,

agricultural water supply, and drinking water supply. Flood prevention and management systems such as a flood forecasting system, flood risk maps, and emergency plans have already been established (Baden-Württemberg, 2014). In recent years, river renaturalization efforts in line with the European Water Framework Directive have called into question if some of these reservoirs should be destroyed.

At the same time, drought events in Germany have been increasing in severity and frequency, including extreme events in

2018 and 2020 (Bundesamt, 2021; Erfurt et al., 2020). The potential shift in annual water availability in the near- and far-future due to both climate and anthropogenic influences (Bundesamt, 2021) is the primary motivator for the state government's development of a 12-point plan for water shortages (Baden-Württemberg, 2021). The 12 actionable points fall under one of five categories: improving monitoring and information, managing and accounting of water uses, strengthening the resilience

of existing water resources, improving awareness and protection incentives, and emergency planning. The potential reuse of flood reservoirs in this state for drought protection could contribute to improved resilience of water resources—provided, of course, that their flood retention capabilities are not impacted.

In this study, we seek to demonstrate the potential water supply benefit of converting pre-existing small (in the global context) flood retention basins into combined flood-drought reservoirs without impacting their flood protection functions. The purpose behind this is twofold: first, to demonstrate that combined flood and drought operation in these small reservoirs is possible in a variety of reservoir sizes; and secondly, to establish a best-case-scenario benchmark for potential combined operation performance. We simulate this operation by modifying the flood operation rules to include drought hedging operations via a retention flow above which the reservoir stores water and a drought threshold target below which we supply water. To maintain flood protection levels, we aim to have a completely empty reservoir before any flood event. The potential water supply benefit of a reservoir is assessed based on the ability of the reservoir to mitigate streamflow drought directly downstream, expressed as a penalty function that more heavily punishes streamflow drought in dry seasons. This is based on the assumption that if the streamflow falls below a seasonal low flow, there is some user (whether anthropological or environmental) that is impacted. We then optimize the retention flow for maximum penalty reduction for a variety of DIN 19700 small, medium, and large flood retention reservoirs in southwest Germany under ideal conditions—that is, with perfect knowledge of the future. We hypothesize that the reservoirs providing the most relative benefit in drought conditions will be those that have high water availability relative to the reservoir capacity. While the limited capacity reduces the reservoir's overall potential benefit, more water available for storage means that a reservoir could potentially store and release its capacity multiple times in a year, increasing the likelihood that it will be able to provide water at critical times.

We begin with a description of the study area and the process of selecting reservoirs for study. Then, we introduce the hydrological model used in this study, as well as the structure of the models representing the current and modified reservoir operations. The modified reservoir also contains two points for hedging: the drought threshold, at which water is released; and the retention flow, for which water is stored and through which the reservoir model is optimized. We then discuss the optimization results (with illustrative examples) and the reservoirs' performance in flood and drought conditions.

## 2    Data and Methods

### 2.1    Study Area

The German state of Baden-Württemberg is in the southwest of Germany and shares borders with France and Switzerland, delineated to the west and south via the Rhine River and Lake Constance. The majority of the state belongs to subcatchments of the Rhine (those of the High Rhine, the Upper Rhine, the Neckar, and the Main tributaries), with the rest belonging to those of the Danube and Tauber catchments.

Two climate regimes dominate, according to the Köppen-Geiger classification (Beck et al., 2023). A temperate oceanic climate (Cfb) covers the majority of the state, including most of the Black Forest and the major cities, such as Karlsruhe, Stuttgart, and

Freiburg im Breisgau. A humid and warm continental climate (Dfb) covers the Swabian Alb and the eastern parts of the Black Forest. Average annual precipitation from 1991-2022 ranges from 600-1200 mm in the majority of the state, though precipitation in the Black Forest is significantly higher (1400-2100 mm). Typical reference evapotranspiration in the same time period ranges from 450 mm per year in the Black Forest and Swabian Alb to 700 mm per year in the Rhine Valley and urban areas.

## 2.2    Reservoir Selection

A subset of potential reservoirs for investigation is first obtained by defining and selecting relevant reservoir categories. Despite the rather large number of very small reservoirs, we exclude these for two reasons: the uncertainties produced when modelling the flows in their small catchments, and the very low expected benefits of their very small capacities. Because we explicitly study the operating rule changes of flood reservoirs, we also exclude reservoirs that do not have flood retention listed as a purpose. We similarly exclude reservoirs with explicit energy production functions, as these typically have strict operating rules that are already optimized, leaving us with two purpose types: flood protection only, or multipurpose with flood protection, where flood protection-only reservoirs tend to have higher flooding thresholds than multipurpose ones. We also distinguish here between reservoirs with permanent and operational inundation: in addition to its potential implications for technical modifications, reservoirs with operational inundation are more likely to have additional complications related to the current land use (e.g. loss of arable land, impacts to reservoir ecosystems). However, because these concerns are not relevant for optimizing water supply, this characteristic is not used in this study but is included for completeness.

The number of representative reservoirs from each category was selected based on a combination of stakeholder interest and representation within the larger subset, with the goal of investigating 30 reservoirs (Table 1). Because the reservoirs in this dual-use scheme are intended to operate independently, there are no constraints relating to spatial connections between reservoirs. Each category containing 15 or more reservoirs was initially assigned three slots (i.e. a reservoir from this category will be chosen) for the reservoir selection. Categories with 40 or more reservoirs were given extra slots depending on the purpose: flood-only reservoirs, which are typically operated in the same manner, were given one extra slot, while multipurpose reservoirs were given two slots due to the variety of uses potentially impacting their operation. After discussion with relevant stakeholders, an additional slot was given to both large categories to allow further investigation of their assumed higher potential. Categories are referred to in this study as a two-letter abbreviation combining their size (where L is large, M is medium, and S is small) and their usage (where F is a flood-only reservoir and M is a multipurpose reservoir). The main categories for this study, their abbreviations, and their distributions (in both the overall reservoir set and the selected subset) can be found in Table 1.

**Table 1. Reservoir categories with abbreviations and number of reservoirs selected for study. Each category abbreviation is a combination of its size (large = L, medium = M, and small = S) and its usage (F = flood-only, M = multipurpose).**

| Size (DIN19700) | Dam Height [m] | Capacity [m³] | Category | Existing Purpose | Inundation Type | # of Reservoirs | # of Selected Reservoirs |
|---|---|---|---|---|---|---|---|
| Large | $\geq 15$ | > 1,000,000 | LF | Flood protection only | Permanent | 6 | - |
| | | | | | Operational | 16 | 4 |
| | | | LM | Multipurpose | Permanent | 26 | 4 |
| | | | | | Operational | 4 | - |
| Medium | 6-15 | 100,000 – 1,000,000 | MF | Flood protection only | Permanent | 18 | 3 |
| | | | | | Operational | 183 | 4 |
| | | | MM | Multipurpose | Permanent | 47 | 5 |
| | | | | | Operational | 17 | 3 |
| Small | 4-6 | 50,000 – 100,000 | SF | Flood protection only | Permanent | 9 | - |
| | | | | | Operational | 128 | 4 |
| | | | SM | Multipurpose | Permanent | 23 | 3 |
| | | | | | Operational | 3 | - |
| Very Small | $\leq 4$ | < 50,000 | VF | Flood protection only | Permanent | 6 | - |
| | | | | | Operational | 143 | - |
| | | | VM | Multipurpose | Permanent | 13 | - |
| | | | | | Operational | 16 | - |

Reservoirs with different degrees of relative water availability from each of the categories were selected to investigate the various hydrological regimes within the region. We define relative water availability here as the availability factor (AF), or

the average number of times per year that a reservoir's capacity (C, in cubic meters) can be filled via the water that we are able to store based on the entire simulation period (excluding the warm-up; i.e. 1998-2021). The water available for storage is the difference between the mean (calculated over the 24 years of simulation) yearly inflow ($Q_{in}$) volume rate and the mean low flow ($Q_{70,mean}$; for definition and calculation see 2.4.1) volume rate, in cubic meters per second (1):

$$AF = \frac{(Q_{in,mean} - Q_{70,mean}) \times 365 \ days}{C} \qquad \text{(1)}$$

The AF can be interpreted as a combined indicator representing the relationship between the water availability in the catchment

and the reservoir's ability to store or release it. A higher AF, then, indicates more water availability relative to the reservoir's capacity. While a reservoir's capacity inherently limits its ability to regulate streamflow, more available water should allow a reservoir to refill more quickly after emptying. In essence, it would increase the likelihood that, in drought conditions, a reservoir would have water to release. This assumption is the basis for our hypothesis that a reservoir with a higher AF should be able to reduce drought conditions more effectively. To test this, we selected reservoirs with varying values of AF, estimated

from local long term statistics (Baden-Württemberg, 2016), from each category. For each combination of category and inundation type, reservoirs whose estimated AFs were close to the 50[th], 25[th], and 75[th] percentile were selected (the distributions of estimated AF for each of the categories can be seen in Appendix A1). A few other reservoirs (Gottswald, Mittleres Kinzigtal, and Fetzachmoos) were selected based on stakeholder interest.

    The resulting 30 reservoirs investigated in this study can be seen in Table 2 and their locations are shown in Figure 1. The

results of the hydrological model LARSIM at these locations were used to then re-calculate the AFs.

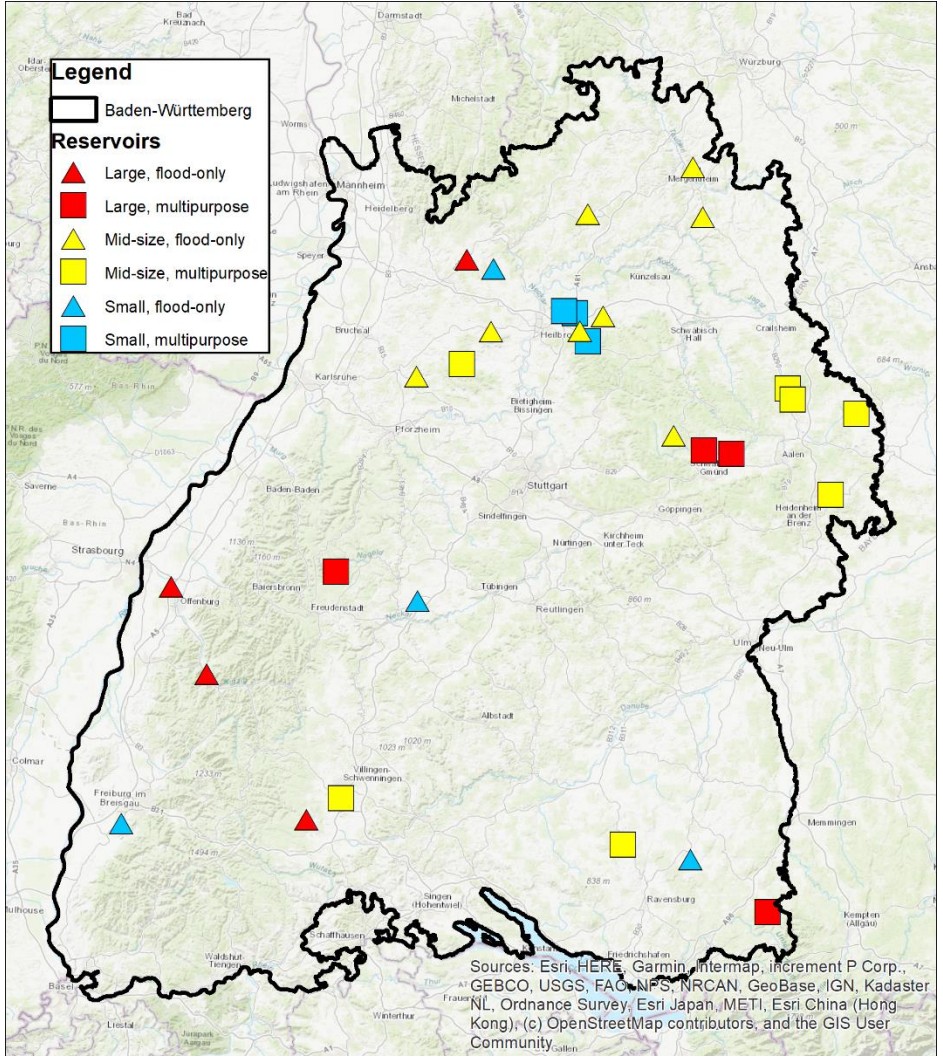

**Figure 1. Locations of selected reservoirs for study in the German state of Baden-Württemberg.**

**Table 2. Selected reservoirs for study, along with their capacities, the flow at which they begin impounding floods (Q$_{crit}$), their operating capacity, their calculated availability factor AF, their catchment area in LARSIM, and the ratio of the actual catchment area to the LARSIM catchment area (used to re-scale the inflow). Note that reservoirs with permanent inundation will have smaller operating capacities than their total capacity.**


| Category | Inundation Type | Name | Operating Capacity [m³] | $Q_{crit}$ [m³s⁻¹] | AF [-] | LARSIM Catchment Area [km²] | Area Ratio [-] |
|---|---|---|---|---|---|---|---|
| LF | Operational | Bernau | 1,020,000 | 22.00 | 17.91 | 112 | 1.00 |
| | | Gottswald | 4,720,000 | 830.00 | 195.90 | 1063 | 1.65 |
| | | Mittleres Kinzigtal | 2,700,000 | 860.00 | 150.1 | 791 | 1.13 |
| | | Wolterdingen | 3,000,000 | 75.00 | 24.56 | 185 | 0.97 |
| LM | Permanent | Federbach | 652,652 | 0.400 | 5.25 | 10 | 1.08 |
| | | Fetzachmoos | 3,500,000 | 15.00 | 17.23 | 4 | 1.04 |
| | | Nagoldtalsperre | 1,741,000 | 15.00 | 5.29 | 39 | 0.93 |
| | | Rehnenmuehle | 2,930,000 | 7.00 | 10.05 | 46 | 0.98 |
| MF | Operational | Schwaigern | 151,880 | 3.32 | 20.41 | 37 | 1.46 |
| | | Seckach | 64,000 | 50.30 | 68.44 | 56 | 0.62 |
| | | Seebaechle | 33,112 | 0.10 | 1.91 | 2 | 1.04 |
| | | Unterbalbach | 210,000 | 6.33 | 14.38 | 29 | 1.24 |
| | Permanent | Doertel | 168,400 | 0.79 | 7.78 | 2 | 0.97 |
| | | Lindelbach | 172,000 | 0.50 | 6.32 | 1 | 1.02 |
| | | Weissacher Tal | 185,000 | 2.41 | 35.16 | 6 | 2.11 |
| MM | Operational | Heinzental | 310,000 | 1.09 | 7.50 | 6 | 1.22 |
| | | Hofwiesen | 335,210 | 10.68 | 91.97 | 26 | 1.02 |
| | | Wustgraben | 276,181 | 0.50 | 6.21 | 6 | 1.00 |
| | Permanent | Fischbach | 181,625 | 3.70 | 11.21 | 16 | 1.00 |
| | | Huettenbuehl | 32,000 | 4.00 | 19.79 | 13 | 1.05 |
| | | Kressbach | 233,780 | 0.70 | 7.09 | 8 | 0.98 |
| | | Michelbach | 81,728 | 1.00 | 4.48 | 5 | 1.16 |
| | | Salinensee | 188,000 | 3.60 | 57.45 | 5 | 1.01 |
| SF | Operational | Duffernbach | 31,143 | 1.55 | 46.70 | 5 | 1.07 |
| | | Goettelfinger Tal | 83,400 | 4.10 | 38.42 | 12 | 1.56 |
| | | Mittelurbach | 60,000 | 0.50 | 22.84 | 7 | 1.02 |
| | | Wollenberg | 30,200 | 3.37 | 36.73 | 8 | 1.55 |
| SM | Permanent | Hoelzern | 7,703 | 1.50 | 5.74 | 1 | 1.00 |
| | | Lennach | 9,600 | 2.10 | 7.37 | 1 | 0.99 |

| | | Nonnenbach | 3,759 | 0.17 | 141.15 | 4 | 1.04 |

### 2.3 Hydrological Model - LARSIM

Semi-natural inflows to each of the 30 reservoirs were calculated using a pre-calibrated version of the Large Area Runoff
Simulation (LARSIM) model (Larsim-Entwicklergemeinschaft, 2023; Ludwig and Bremicker, 2006), provided by the State
Agency for the Environment of Baden-Württemberg (Landesanstalt für Umwelt Baden-Württemberg, LUBW). LARSIM is a
process-based water balance model that can be either semi- or fully distributed, and takes as inputs geographic data (elevation,
land use, and soil parameters) and hydrometeorological data (precipitation, air temperature, humidity, windspeed, radiation,
and water temperature) to provide operational streamflow forecasts in the region.

The model uses a grid structure with a 1 km$^2$ resolution to describe meso-scale hydrological processes such as interception,
evaporation using the Penman-Monteith method, snow-related processes (accumulation, compaction, and melt), river routing,
and soil water storage to evaluate discharge and water temperature. Thus, while typically used for large catchments (and
calibrated to higher-order river discharges), it is also capable of modelling smaller headwater catchments by selecting the
proper model output location. These model output locations were selected to have LARSIM-delineated catchments that are as
similar as possible to actual conditions (e.g. connecting tributaries, catchment area). However, due to the 1 km$^2$ grid and
different channel routing procedures, the LARSIM catchment area may differ from the true catchment area. We adjust for this
by multiplying the resulting discharge by the ratio of true catchment area to LARSIM catchment area (the exception here being
Fetzachmoos, whose main structure as a diversion dam is not in the river network and whose delineated catchment area does
not model the water that should be impounded).

We refer to the resulting discharges as semi-natural because the model also incorporates anthropogenic influences such as
operations of water treatment plants and selected reservoirs and dams (Baden-Württemberg, 2024b): if a selected reservoir is
upstream of another selected reservoir, we include the current calibrated operations of the upstream reservoir for the inflow to
the downstream. The provided model includes data from over 265 discharge gauges and 390 precipitation gauges, as well as
hundreds of available meteorological stations, and is currently used in flood forecasting operations by the state agency. Its
output contains 24 years of hourly data from 1997-2021 (however, two reservoirs, Gottswald and Mittleres Kinzigtal, have
only 23 years of available data).

### 2.4 Reservoir Models

Two reservoir operation models were programmed: one modeling the current operation (i.e. the flood-optimized condition),
and one modeling the potential combined (i.e. flood and drought) operation.

**The flood operation model** consists of three modules: flood operation, in which discharge above the flooding limit
downstream ($Q_{crit}$) is stored until the reservoir's operating capacity is reached; flood release, which empties the reservoir once
the flood wave passes; and normal operation, in which there is no change to the reservoir's volume. $Q_{crit}$ is the design flood for

the reservoir; if there are urban areas downstream, this is typically the 100-year flood . If the reservoir is full before the flood wave passes, the additional water is returned to the river channel and is considered flood failure. This is a generalized version of the current reservoir operation rules for all selected reservoirs, regardless of existing uses. (In the interest of completeness, we note that some of these reservoirs have seasonally variable operational capacities—this variation has been ignored in this study.)

**The combined operation model** expands on the flood operation model in three ways:

1. The reservoir releases water for drought once the discharge has fallen below a certain drought-related threshold (in this study, we use the 70[th] percentile exceedance flow; for calculation, see 2.4.1);
2. To increase water available for drought releases, the model introduces a retention flow ($Q_r$) above which the reservoir impounds water (when $Q_{in} > Q_r$, the reservoir stores $Q_{in} - Q_r$). This is the variable parameter through which we optimize the model; and
3. Instead of releasing the retained flood volume immediately after the flood wave passes, the reservoir holds onto the water until the drought threshold is met (in which case it releases the water) or another flood wave is predicted. The forecast horizon for the flood wave in this perfect-knowledge scenario is the drawdown time, or the time the reservoir needs to empty the current volume. If a flood wave does occur, the reservoir empties its contents and remains empty (i.e. ignores the $Q_r$ filling condition) until the flood wave begins. In this way, we ensure that the flood retention capability of the reservoir is not compromised.

This model only requires an inflow time series, a flooding limit, and the reservoir capacity, and produces a drought threshold time series that is used to calculate a volume time series, an outflow time series, and a penalty time series, which seeks to evaluate the reservoir's performance (for calculation and explanation, see 2.4.3). Because these inputs are often relatively accessible, this model is rather flexible and can be applied to many reservoirs, even those outside of Baden-Württemberg. A flow chart of the combined operation model can be seen in Figure 2.

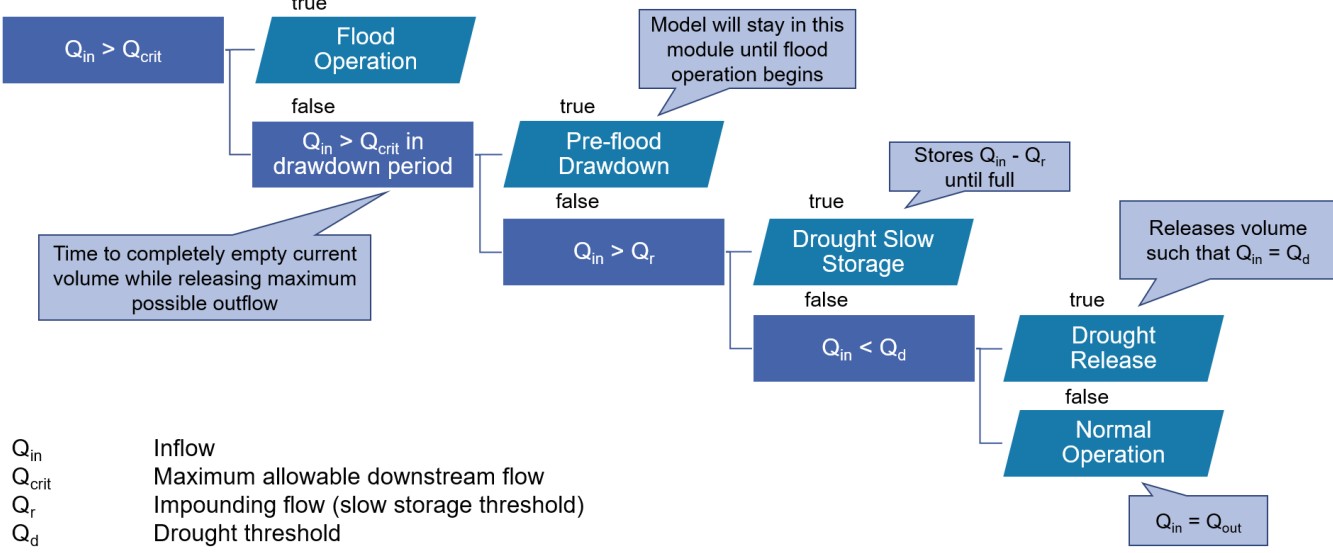

**Figure 2. Decision tree for combined operation model.**

### 2.4.1 Drought Release Targets

Drought remains a complex and multivariate phenomenon that affects multiple sectors, though different users may experience these effects at different times. This makes drought difficult to quantify and define. A distinction should be drawn between drought events and drought conditions: drought conditions are levels of intense dryness below a certain threshold, whereas drought events are prolonged periods of drought conditions (usually with a minimum duration of 30 days). Given that a reservoir's most immediate impact is on streamflow, we focus here on its potential ability to decrease streamflow drought conditions via streamflow drought thresholds as a preliminary step into its ability to reduce drought. A truly comprehensive drought reduction approach would not just consider hydrologic variables but also consider management techniques (which is beyond the scope of this paper) for soil moisture, agricultural, and ecological drought within a given catchment to manage the prolonged dryness. However, hydrological droughts still have implications for impacts on other sectors, such as reduced drinking water or irrigation water availability (Van Loon, 2015), and many healthy ecosystems depend on certain flows at certain times (Yarnell et al., 2020). Streamflow drought, often expressed as a threshold level, is a common hydrological drought indicator.

Because such thresholds can be extremely variable and location-specific, especially for reservoir flows, a method that could be applied to different 30 reservoirs was needed. This method should also allow for seasonal variability, as previous studies on reservoir hedging rules for preventing drought have demonstrated that such rules are most effective when allowed to vary throughout the year (Chang et al., 1995; Balley, 1997). The drought release targets in this study should therefore be a streamflow drought threshold that allows for seasonal variability that could be applied anywhere.

The drought threshold used in this study is the percentile exceedance flow per Cammalleri et al. (2016), with a minor adjustment for the hourly time step of the model output. For each time step $t$ within a year, we collect a 721 x $n$ matrix of discharge values: 721 represents all the hourly time steps in a 30-day moving window (with an additional value to center the window on $t$), which is applied to all the years in the dataset ($n$). The cumulative distribution function curves for discharge, and then the percentile exceedance curves, are derived based on the values in this matrix. The threshold value at each timestep is the discharge corresponding to the chosen percentile exceedance. This means that at the 70[th] percentile, this threshold discharge is exceeded 70% of the time. Typical reference values in the literature range from the 70-95[th] percentile (Hisdal et al., 2004; Cammalleri et al., 2016; Van Loon et al., 2010) and are generally adjusted for river dynamics (flashier rivers, for example, would typically select a higher percentile)—though in the interest of consistent inter-reservoir comparisons, we will choose a singular percentile threshold for all reservoirs. Then the steps are repeated until there is a value for each hourly time step in a year. The result is a seasonally-variable drought threshold for that time step that is dependent only on a streamflow time series, making it easily applicable to different locations. This calculation is summarized in Figure 3.

The choice of percentile has a notable effect on the detection of streamflow droughts: a higher percentile exceedance generally means more intense drought conditions and fewer drought events (Cammalleri et al., 2016; Tallaksen et al., 2009). We use the 70[th] percentile exceedance flow ($Q_{70}$) as the threshold here, as it is the most lenient definition among typical values and allows

insight on the reservoirs' ability to mitigate not only severe drought conditions but also mild ones. If the inflow at any time step is less than $Q_{70}$ (i.e. the discharge drops below the threshold), we assume that there is some user of the water—whether human or otherwise—that is being impacted by water scarcity in the river. To mitigate these consequences, the combined model uses stored volume in the reservoir (if any) to supplement the outflow such that $Q_{70}$ is reached. This low threshold will allow us to evaluate the new rules' ability to alleviate both mild and severe droughts. The arithmetic mean of this $Q_{70}$ time

series is also used as an estimate of the average low flow to calculate the AF in Eq. (1).

The exceedance percentile flow is commonly used to represent streamflow drought conditions when defining drought events (Van Loon et al., 2010; Van Loon and Van Lanen, 2012; Hisdal et al., 2004; Tallaksen et al., 2009; Cammalleri et al., 2016) and is currently used (at the 75[th] percentile) as a warning indicator for low flow in Baden-Württemberg (Baden-Württemberg, 2024a). The percentile exceedance has also been used in studies seeking to define ecological flows, though usually at the 85[th]

percentile (Knight et al., 2011; Knight et al., 2013; Vigiak et al., 2018; Yarnell et al., 2020). The 85[th] percentile may also serve as a regulated lower limit for agricultural water abstraction, as noted in Salmoral et al. (2019).

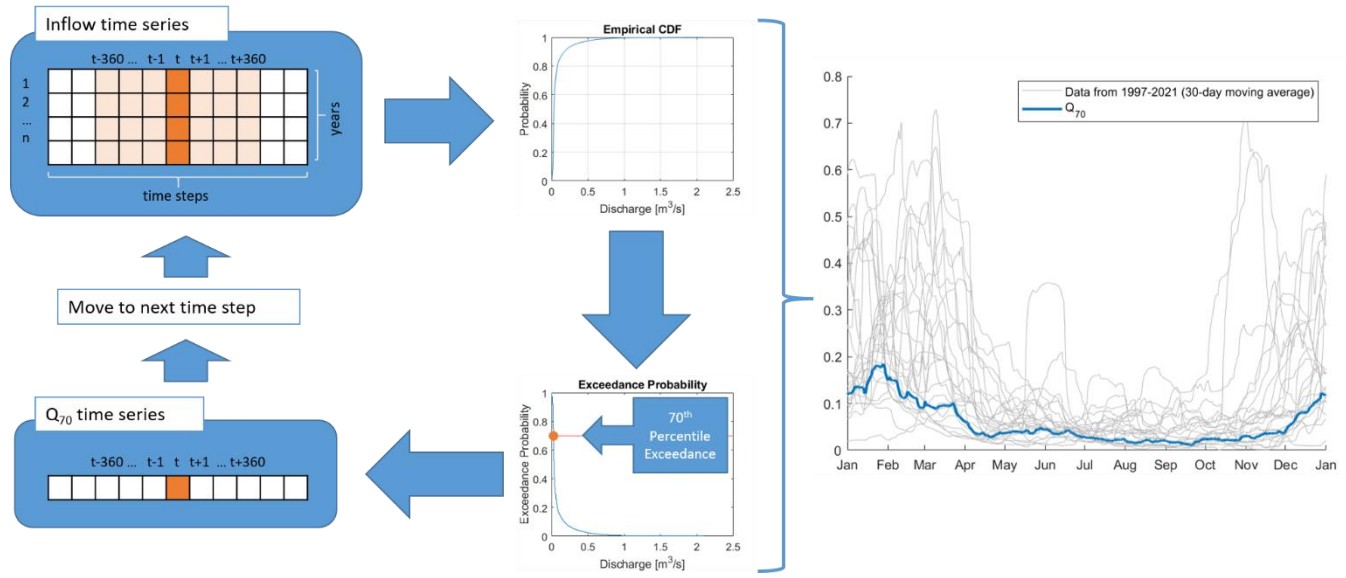

Figure 3. Example calculation of the $Q_{70}$ time series.

The hourly resolution of this study's demand time series may be difficult to use in practice: reservoirs typically change their releases on weekly or monthly scales. We retain this high temporal resolution, however, to match the hourly resolution of flood forecasts and operations. In the future, known thresholds or demand curves (derived e.g. from local irrigation demands) may be substituted for the percentile exceedance curve.

### 2.4.2 Pre-Flood Drawdown Time

The combined operation model was programmed with the assumption of perfect knowledge of inflow and in particular of flood onsets. In practice, this means the forecasting horizon ($t_{down}$) should be calculated for every non-flood time step. The forecasting

horizon is the time $t_{down}$ such that the potential release from the reservoir is greater than or equal to the volume at the end of the current time step (2):

$$\int_{t_i+1}^{t_i+t_{down}} [Q_{crit} - Q_{in}(t)]\, dt \geq V(t_i) \tag{2}$$

After calculating $t_{down}$, the model checks if a flood begins ($Q_{in} > Q_{crit}$) within the next $t_{down}$ timesteps. This is, in effect, a perfect-knowledge flood forecast. If there is a flood, the model enters the pre-flood drawdown module in which the reservoir is emptied by releasing the water at $Q_{crit}$. Once emptied, the reservoir remains empty until the flood event begins. By ensuring that the flood reservoir is empty before onset, we guarantee that the flood protection function is not compromised.

### 2.4.3   Expressing Degrees of Reservoir Failure

Degrees of reservoir failure (i.e. excess discharge above $Q_{crit}$ and deficit discharge below $Q_{70}$) in both flood and drought at each time step are expressed in this model as penalties. Flood ($P_f$) and drought penalties ($P_d$) calculated using the flood operation model are considered the baseline penalties for each reservoir and are handled as separate time series. The penalties serve three functions in this study:

1.  To evaluate the preservation of flood protection during the optimization phase. The flood penalty in the flood operation model ($P_{f,f}$) is used as the baseline standard—if the flood penalty of a combined model run ($P_{f,c}$) shows a higher penalty than the $P_{f,f}$ at any time step, the solution is rejected.

2.   To assign hypothetical "damages" to reservoir failure in both drought and flood. Flooding volume should always be strongly penalized; however, assigning a flat value to all flood volumes is not ideal because it will be unable to capture increases in flood volumes. In drought failures, greater water deficits should be more heavily penalized than smaller ones.

3.  To evaluate the effect of the changes to operating rules by comparing the reduction in "damages" from the optimized models.

As with the drought threshold definitions, these penalty functions can be replaced with a different method of expressing degrees of failure as a function of discharge or height, if a river rating curve exists (e.g. monetary flood damage per unit excess discharge).

Because the flood penalty at time t ($P_{f,t}$) is only used to ensure flooding does not increase, a simple calculation is desired. Moreover, no penalty should be given if the reservoir outflow is less than or equal to the downstream flooding discharge. Here, it is a linear transformation (arbitrarily given a slope of 5) of flooding downstream of the river where penalty increases significantly once the outflow $Q_{out,t}$ exceeds the flooding discharge ($Q_{crit}$) (3):

$$P_{f,t} = \begin{cases} 0, & Q_{out,t} \leq Q_{crit} \\ -5(Q_{out,t} - Q_{crit}), & Q_{out,t} > Q_{crit} \end{cases} \tag{3}$$

The drought penalty functions at time t ($P_{d,t}$) are selected based on the assumption that small deviations of $Q_{out}$ from the drought threshold $Q_{70}$ will be less impactful (and therefore less penalized), while also strongly penalizing outflows closer to zero. For this, we chose a square root function, which penalizes small deviations lightly but increases exponentially as the discharge approaches zero. Penalties for $Q_{out}$ below 0.00001 m$^3$s$^{-1}$ are assumed to be the same as for $Q_{out}$ of 0.00001 m$^3$ s$^{-1}$ to avoid potential division by infinity. This results in the following penalty expressions (4):

$$P_{d,t} = \begin{cases} 0, & Q_{out,t} \geq Q_{70,t} \\ -\dfrac{1}{\sqrt{Q_{out,t}}} + \dfrac{1}{\sqrt{Q_{70,t}}}, & Q_{out,t} < Q_{70,t} \\ -\dfrac{1}{\sqrt{0.00001}} + \dfrac{1}{\sqrt{Q_{70,t}}}, & Q_{out,t} \leq 0.00001 \end{cases} \tag{4}$$

Because penalty at $t$ is a function of the seasonally variable $Q_{70}$ at that time step, penalty also has an element of seasonality. The penalty per missing unit volume of water changes with $Q_{70}$: it will be higher in seasons where $Q_{70}$ (and therefore streamflow in general) is low, and lower in seasons where $Q_{70}$ is high. For example, the penalty of missing 1 m$^3$ s$^{-1}$ if $Q_{70}$ is 2 m$^3$ s$^{-1}$ is -0.293; if $Q_{70}$ is 10 m$^3$ s$^{-1}$, the penalty is -0.0171. In this way, the model correctly penalizes shortages in the dry season more heavily than during the wet season.

For discussion of results between reservoirs, we evaluate the reduction of penalty and drought deficit volume between the combined operation model and the flood operation model. These comparisons are done without results from the first year of operation to allow for a warm-up period.

We express the penalty benefit for drought ($B_p$) as the percent reduction in total drought penalty from the flood operation model ($P_{d,f}$) in comparison to that of the combined operation model ($P_{d,c}$), normalized by the $P_{d,f}$: (5)

$$B_p = 100 \times \frac{\sum P_{d,f} - \sum P_{d,c}}{\sum P_{d,f}} \tag{5}$$

Because penalty has an element of seasonality, the benefit per unit volume of water is also seasonal: the benefit associated with providing 1 m$^3$ s$^{-1}$ will be higher when $Q_{70}$ is low than when $Q_{70}$ is high.

We similarly describe the volume benefit for drought ($B_v$) as the percent reduction from the total drought deficit volume of the flood operation model ($V_{d,f}$) in comparison to that of the combined operation model ($V_{d,c}$), normalized by the $V_{d,f}$:

$$B_v = 100 \times \frac{\sum V_{d,f} - \sum V_{d,c}}{\sum V_{d,f}} \tag{6}$$

The volume benefit $B_v$ differs slightly from the penalty benefit $B_p$ in that penalty allows heavier weighting of water delivery at critical times: the same volume of water may reduce penalty by different amounts, depending on the season. In contrast, the volume benefit assumes that every unit of water is equally valuable, regardless of when it is delivered.

The total volume released by the reservoir for drought protection purposes ($V_d$) is normalized by the reservoir capacity (C):

$$V_{d,nor} = \frac{V_d}{C} \tag{7}$$

Thus, $V_{d,nor}$ indicates the number of times the reservoir's complete capacity is given for drought protection over the model simulation.

### 2.4.4 Optimization of Retention Flow for Drought Mitigation

The reservoir model was programmed with the following constraints:

- The reservoir volume at the end of time t ($V_t$) is equal to the volume at t-1 plus the difference between the inflow ($Q_{in,t}$) and outflow ($Q_{out,t}$) at time t (8);

- The operating capacity C is the operational volume of the reservoir; in other words, the difference between the full reservoir volume and the permanent inundation volume (which, for operationally-inundated reservoirs, is zero) (9);

- The reservoir volume cannot exceed the operating capacity and cannot be less than 0 (10);

- The reservoir outflow at time t ($Q_{out,t}$) is dependent on the current volume and the inflow. Moreover, $Q_{out,t}$ can only exceed $Q_{crit}$ in a flood failure scenario (i.e. normal releases cannot exceed $Q_{crit}$) (11); and

- The retention flow Qr must be between the highest value in the release target time series and the flooding limit (12)

$$V_t = V_{t-1} + (Q_{in,t} - Q_{out,t}) \times t \tag{8}$$

$$C = V_{full} - V_{permanent\ inundation} \tag{9}$$

$$0 \leq V_t \leq C \tag{10}$$

$$Q_{out,t} = \begin{cases} Q_{out,t} \leq Q_{crit}, & V < C \\ Q_{out,t} > Q_{crit}, & V - C > 0 \end{cases} \tag{11}$$

$$\max(Q_{70}) < Q_r < Q_{crit} \tag{12}$$

The constraint on the retention flow $Q_r$ comes from the logic of the reservoir operation. If the inflow to the reservoir exceeds $Q_{crit}$, the reservoir will already be retaining water; thus, $Q_r$ must be less than $Q_{crit}$ in order to allow storage of non-flood water. A lower $Q_r$, then, is more likely to increase total water storage for drought but will have no effect on flood protection. If the inflow to the reservoir is below $Q_{70}$, the reservoir will release stored volume to increase the outflow to the threshold. Each reservoir under the combined operation model was simply optimized by testing 50 equidistant values of $Q_r$ between the

maximum of $Q_{70}$ and $Q_{crit}$ to cover the range of possible values. The resulting $P_f$ and $P_d$ were used to evaluate the run: any $Q_r$ that resulted in an increase of $P_f$ was excluded from simulation, and the $Q_r$ that resulted in the lowest drought penalty (i.e. highest benefit) was considered the optimal $Q_r$ for the reservoir.

## 3 Results & Discussion

### 3.1 Optimization Results

The reservoirs' $Q_{crit}$, maximum and minimum $Q_{70}$, and the optimal $Q_r$ are available in Appendix B1.

To evaluate the reservoirs' effectiveness in reducing drought conditions, we plot the penalty benefit (which is a function of both drought time and deficit) against the availability factor for all reservoirs in Figure 4, yielding interesting results. It seems that reservoirs with high flood pre-release volumes and lower AF (such as Rehnenmuehle, Fetzachmoos, Heinzental, and Federbach) still were able to reduce penalty significantly: this is interesting, as it implies that reservoirs with less water available can still refill quickly enough after a flood event to supply water during drought deficits.

Indeed, contrary to our hypothesis, reservoirs with high AF were unable to decrease penalty significantly. It seems that overall, there is an inverse relationship between benefit and availability in the large and mid-size categories, with small reservoirs being standout exceptions. Overall, LM, MF, and MM reservoirs were the most effective at reducing drought: all LM reservoirs had penalty benefits of over 70%, while more than half of MM and MF reservoirs had benefits of over 50%. Our sample of SF reservoirs as a category outperformed LF reservoirs, indicating that even small reservoirs can have noticeable benefits for local streamflow drought reduction.

SM reservoirs, however, are almost completely ineffective. In the cases of Hoelzern and Lennach, it is because the reservoirs had a $Q_{crit}$ that was so high in comparison to the modeled inflow—in the 24 years of simulation, they were generally on the scale of 0.005 $m^3$ $s^{-1}$ and did not experience a single flood wave large enough to impound water. Even the lowest $Q_r$ value tested barely allowed for any storage of water in the combined scenario. Thus, there was almost no water available to release. Even when there was water available, as in the case of Nonnenbach, there was simply not enough volume in the reservoir to compensate for the drought deficits. Because of this, we consider these reservoirs generally unsuitable for a combined use strategy under these conditions.

The reservoirs can be roughly grouped into one of five groups based on Figure 4: the two SM reservoirs with almost no benefit; low-availability (< 100 AF), high benefit (> 60%) reservoirs; reservoirs with middling benefit (50-60%); reservoirs with low availability and low benefit (< 45%); and high-availability (> 100 AF) reservoirs with low benefit. In the next sections, we explore the combined model outputs of selected reservoirs (shown in Table 3) from the non-SM groupings to understand the interactions of AF, benefit, and release volume.

**Table 3. Selected reservoirs for exploration.**

| Reservoir Name | Category | AF [-] | Normalized Release Volume [-] | Benefit [%] |
|---|---|---|---|---|
| Gottswald | LF | 195.90 | 46.90 | 10.35 |
| Heinzental | MM | 7.50 | 5.85 | 96.87 |
| Hofwiesen | MM | 91.97 | 28.68 | 50.64 |
| Federbach | LM | 5.25 | 2.65 | 63.91 |
| Wollenberg | SF | 36.73 | 21.58 | 37.78 |

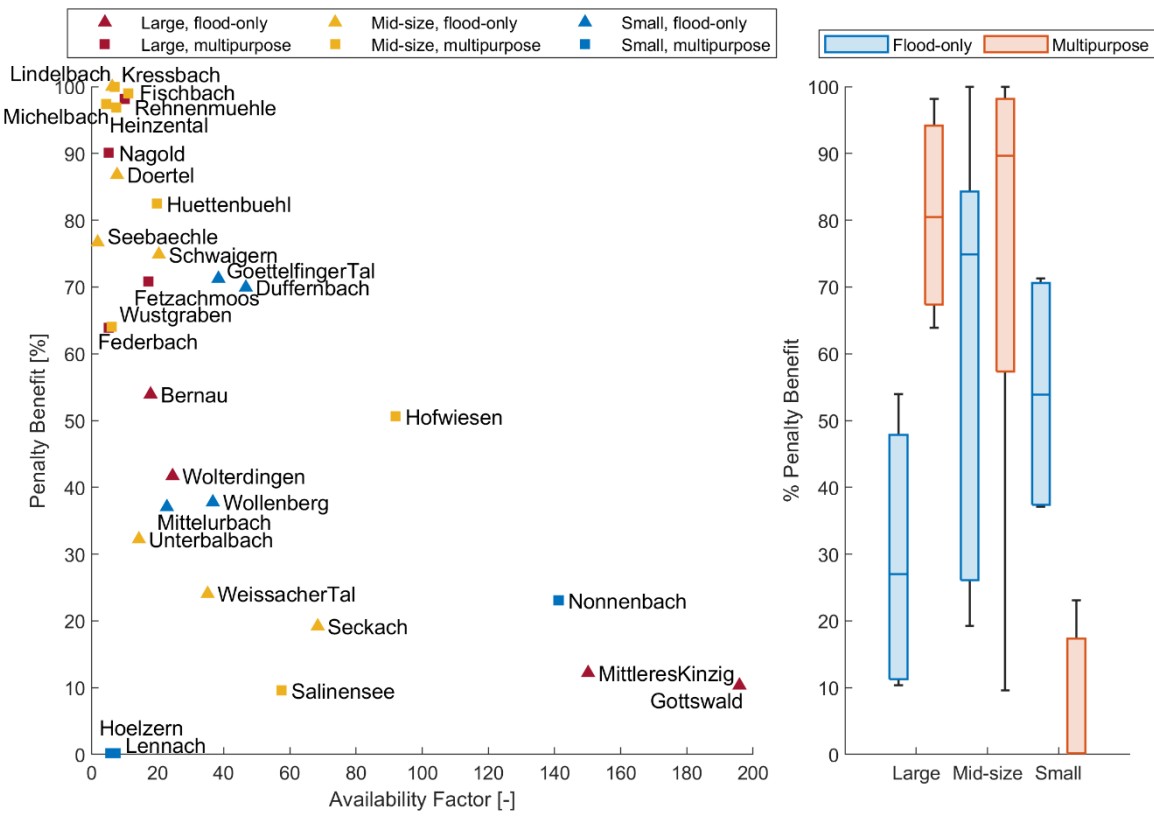

**Figure 4. Penalty Benefit relative to availability factor (left) and summarized by reservoir size and use categories (right).**

### 3.1.1    High Availability, Low Benefit - Gottswald

Gottswald is a large flood-only reservoir with very high relative water availability—the highest of all the selected reservoirs—but is only able to reduce roughly 11% of the total penalty. Investigation into the discharge, volume, and penalty time series (Figure 5) shows that while high discharge events are common, strong drought penalties are also common and long-lasting. Because no flood waves greater than $Q_{crit}$ occur within the simulation years, there is no pre-flood release from the reservoir and all water released is for the purpose of drought protection. The reservoir—as a result of the introduction of $Q_r$—is able to

store and release significant amounts of water, as one would expect of a location with high relative water availability. However, even when filled to its capacity, the reservoir is unable to release enough water to overcome anything beyond the mildest drought peaks, often reaching zero before the drought conditions intensify. Even deficits with relatively small penalties such as those in October 2009 and January 2010 (see Figure 6) are quite substantial, with deficits of up 5 m$^3$ s$^{-1}$. The reservoir at full capacity (2.7 million m$^3$) can only sustain this deficit for just over six days. Thus, while the reservoir's current capacity is

capable of supplementing water for short periods of time, the deficit volume is simply too big in comparison.

This presents a problem with our hypothesis of AF as an indicator for penalty reduction. A large AF per our definition would indicate either a very small volume relative to the typical catchment flows or a very strongly variable catchment flow. The discharge time series in Figure 5 suggests that it is the latter—indeed, the discharge time series shows extremely strong peaks that are over 100 times the $Q_r$ which fill the reservoir quite quickly, while drought deficits drain the water almost as quickly.

The reservoir volume is simply too small to take advantage of the available water. At the same time, the large deficits are (at least in part) a result of the $Q_{70}$ as the drought definition: in a highly variable flow regime, this lenient definition may select flows that are unrealistically high for dry conditions. In these cases, it may be more realistic to choose a higher percentile exceedance for a more optimized operation. However, we retain the use of $Q_{70}$ so that the operation of different reservoirs in this study are analyzed at the same relative thresholds.

### 3.1.2    Low Availability, High Benefit - Heinzental

Heinzental (Figure **7**) is a mid-size multipurpose reservoir with a low AF but significant drought improvements. Flood events occur several times throughout the time series; however, it is able to completely protect against flood events while still being able to compensate for the vast majority of drought conditions. In contrast to Gottswald, Heinzental requires significantly less water to overcome the drought conditions at the inlet, often completely overcoming the penalty conditions entirely before even

half the volume is used. The only times the reservoir fails to overcome drought are following a sharp intensification of drought conditions immediately after a flood event in 2011, and in 2017 after compensating for another intensification of drought conditions. This seems to be due to the very stringent drought threshold: the maximum value in the threshold time series is 0.05 $m^3$ $s^{-1}$. Even if upstream the river were dry (i.e. no inflow to the reservoir), the reservoir's capacity could supply that discharge for 54 days. It is likely that any further changes to the reservoir's rules could improve the efficiency of such

reservoirs, as it is already quite high.

### 3.1.3    Low Availability, High Benefit – Federbach

Federbach (Figure 8) is a large multipurpose reservoir with a rather low AF in comparison to other large reservoirs. As it is on the lower end of the high-benefit reservoirs, it demonstrates some limitations that impact a reservoir's benefit. It frequently impounds flood volumes—this means that much of the stored volume is released not for drought protection but to ensure an

empty reservoir for flood protection. Unfortunately, the reservoir fails in a couple of flood events; however, because the reservoir in the flood-only operation also could not completely retain these events, these do not represent an increase in flood risk. Additionally, the reservoir often struggles to reach full capacity (roughly 652,000 $m^3$) due to the frequent flood pre-releases, as the flood waves are often not enough to fill the reservoir completely. Despite this, the reservoir does manage to eliminate many of the smaller drought penalty events. Assuming the reservoir needed to supplement the maximum $Q_{70}$ of

0.0932 $m^3$ $s^{-1}$ to a dry riverbed, Federbach's capacity could last for almost 81 days. In this sense, it is the opposite of Gottswald—a reservoir with a capacity that is more than capable of delivering the needed water, making its benefit quite high. However, its potential for drought alleviation is limited by the frequency of floods.

### 3.1.4 Middling Benefit – Hofwiesen

Hofwiesen (Figure 9) is a mid-size multipurpose reservoir with the highest AF of all selected mid-size reservoirs. The reservoir is able to compensate for a lot of drought penalties with its capacity and, in line with our hypothesis, its high water availability allows it to refill quickly, allowing it to give more water in drought conditions. However, the reservoir fails with strong and prolonged drought signals, such as those extending from 2011 through 2012 and from 2018 through 2020, despite being able to refill a couple of times. Bernau, which is the other reservoir with middling benefit, shows similar behavior. Overall, this reservoir grouping is capable of dealing with smaller, shorter drought conditions in the river.

### 3.1.5 Low Availability, Low Benefit - Wollenberg

Wollenberg (Figure 10) is a small flood-only reservoir. In addition to having a low AF among small reservoirs, it also has the lowest improvement of all reservoirs. As with many reservoirs in this grouping, it is rather clear that the low benefit comes from a lack of water: the reservoir is only able to fill a few times, in part because the reservoir never experiences any floods. Indeed, the flooding limit is more than 10 times the highest discharge. This is an indication that the reservoir could be overbuilt: in other words, $Q_{crit}$ is too large in comparison to the average flow. In our simplified optimization process where we test 50 evenly spaced values between $Q_{crit}$ and the drought threshold, this results in a $Q_r$ that never allows the reservoir to completely fill. Even when $Q_r$ is reached, the reservoir only reaches 1/3 of its usable capacity (30,200 $m^3$). With the reservoir levels so low most of the time, the reservoir can hardly compensate for any drought events. It seems likely that further decreasing $Q_r$ would significantly increase the volume of stored water and possibly the benefit.

Such a solution poses another general question—how far should the $Q_r$ be lowered? It seems that, given perfect knowledge, it should be possible to lower $Q_r$ to the drought threshold. However, this could result in significant and potentially catastrophic changes to the river regime. For example, aquatic species that require moderate flooding from time to time could be severely affected by the attenuated discharges from a much lower $Q_r$. At the same time, a highly regulated river regime could be beneficial for agricultural planning or industry. Because our study focuses on the general benefits of reservoirs for water supply without making assumptions about the uses downstream, these questions are ultimately outside the scope of this paper but should be considered for future studies.

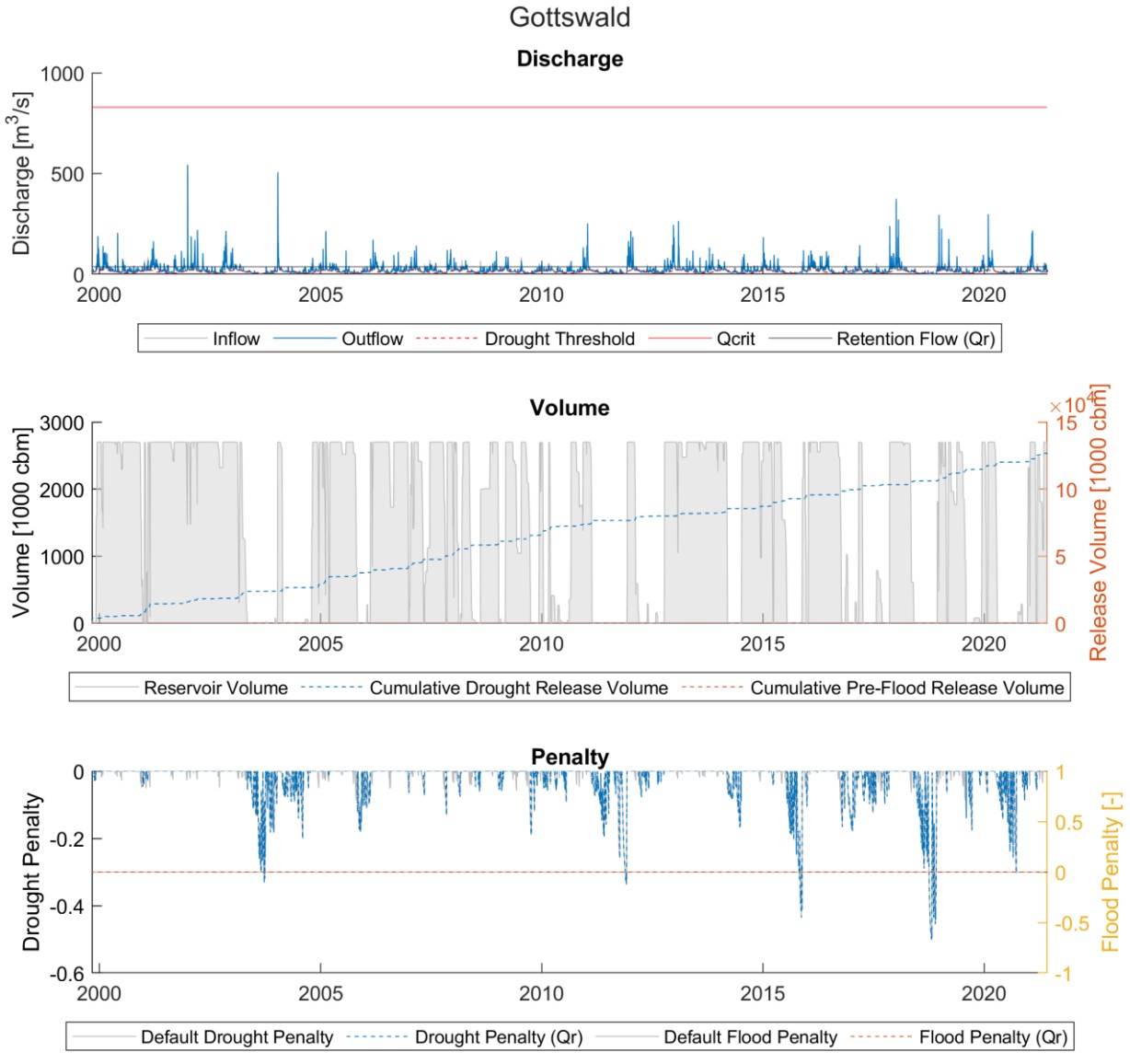

**Figure 5. Discharge, volume, and penalty time series for Gottswald reservoir (example of a high availability, low benefit reservoir).**

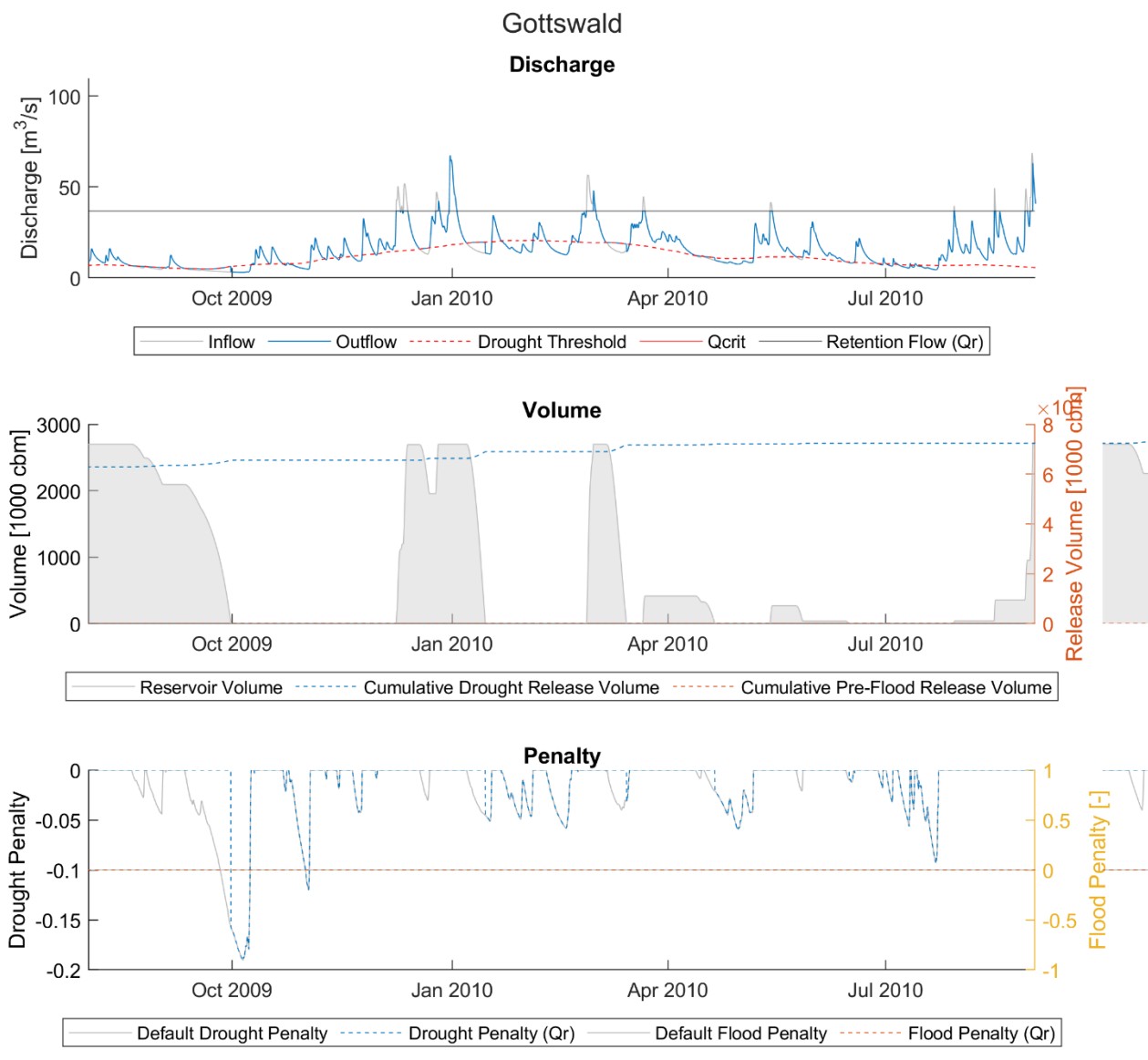

**Figure 6. A closer look at a problematic period for Gottswald reservoir.**

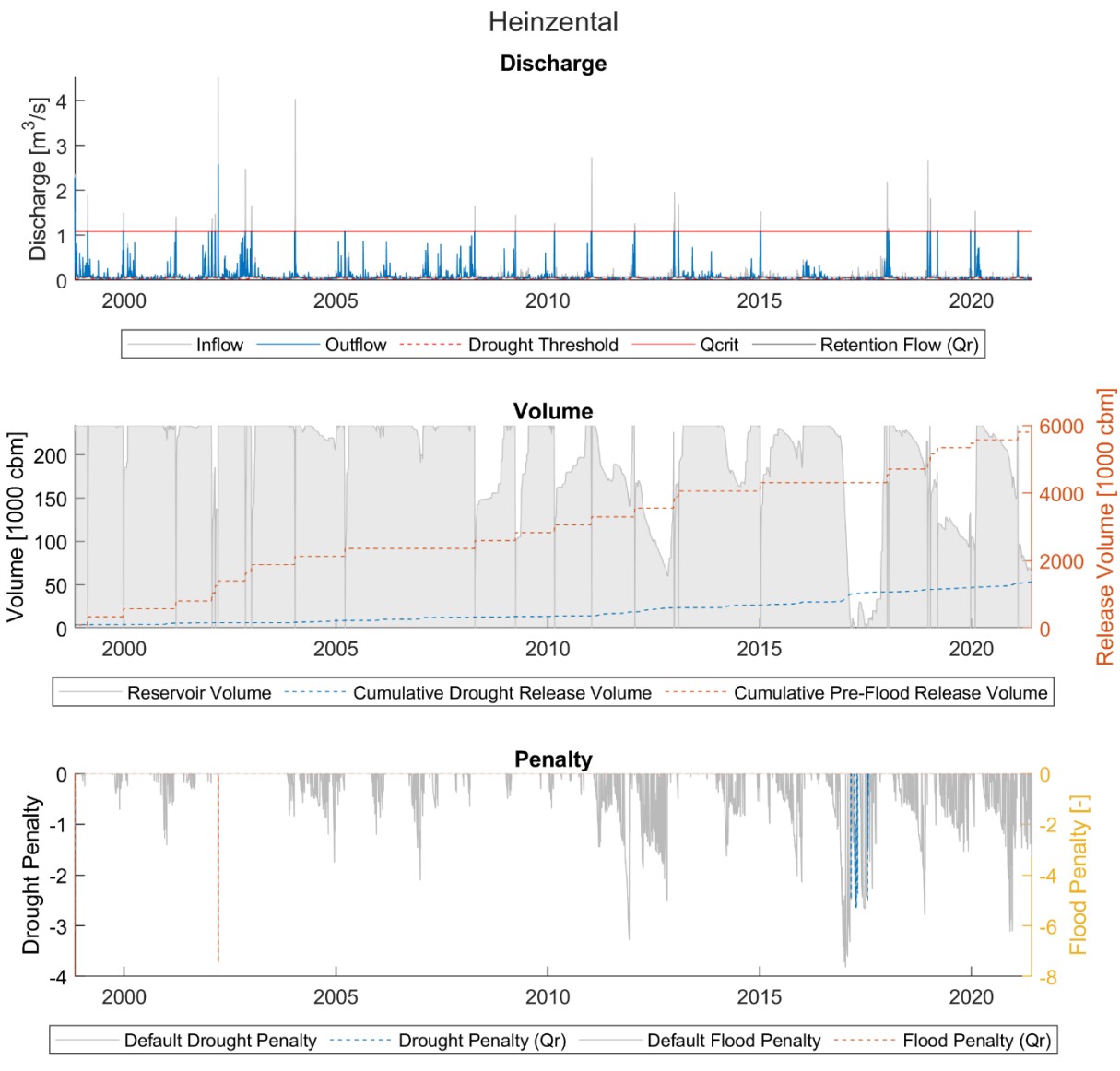

**Figure** 7**. Discharge, volume, and penalty series for Heinzental reservoir (example of a low availability, high benefit reservoir).**

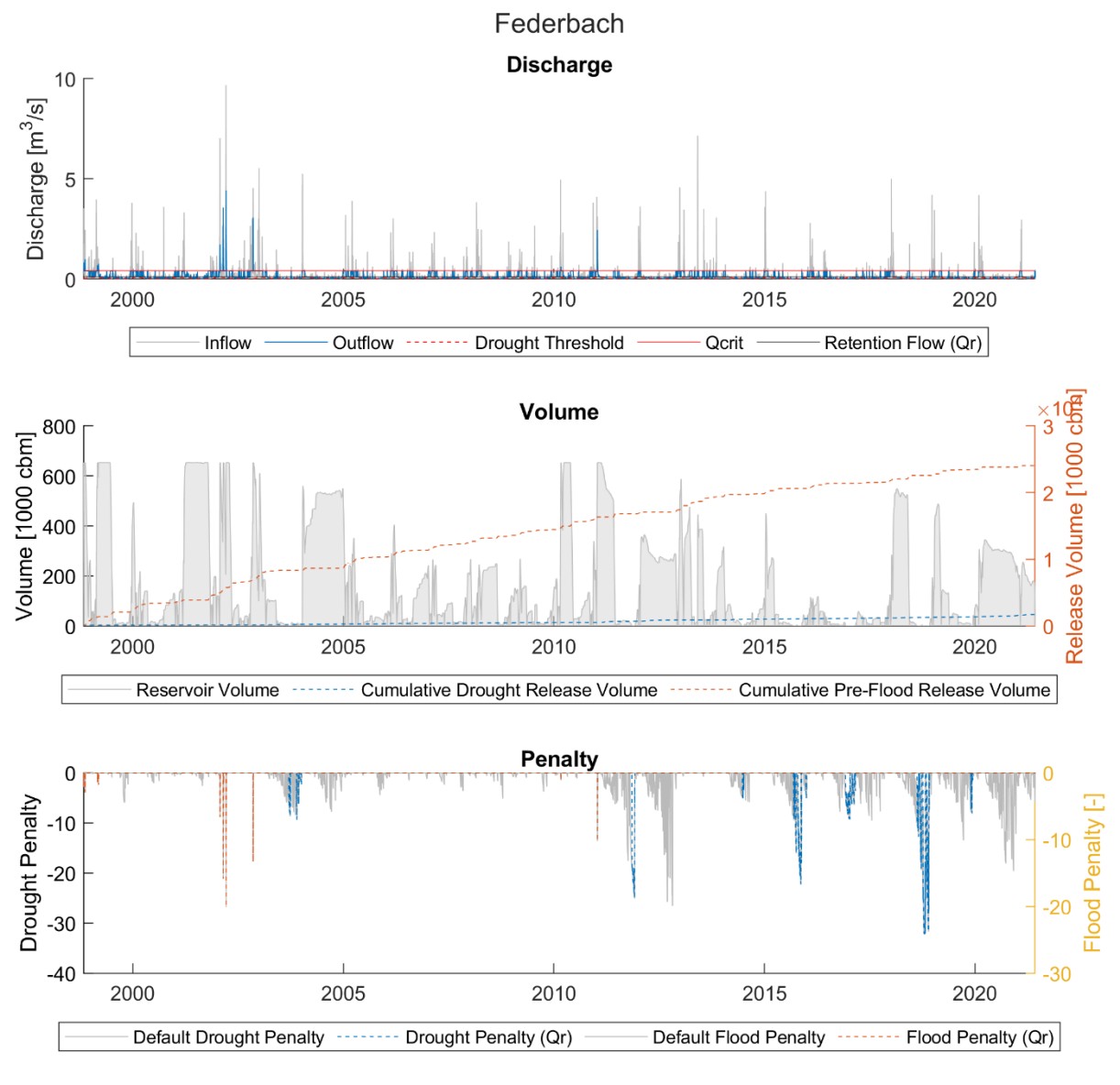

**Figure 8. Discharge, volume, and penalty time series for Federbach reservoir (example of a low availability, high benefit reservoir).**

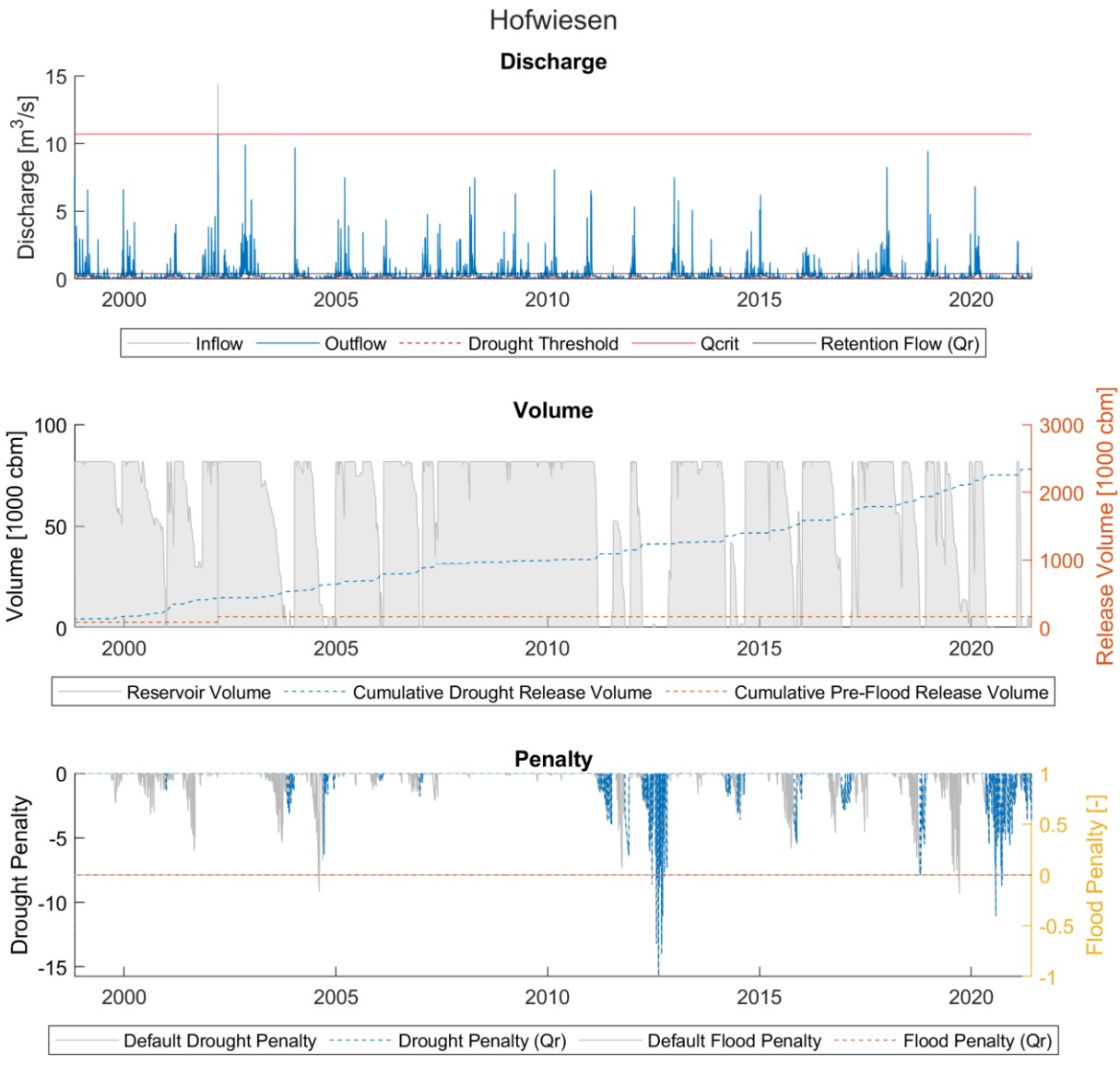

**Figure 9. Discharge, volume, and penalty time series for Hofwiesen reservoir (example of a middling-benefit reservoir).**

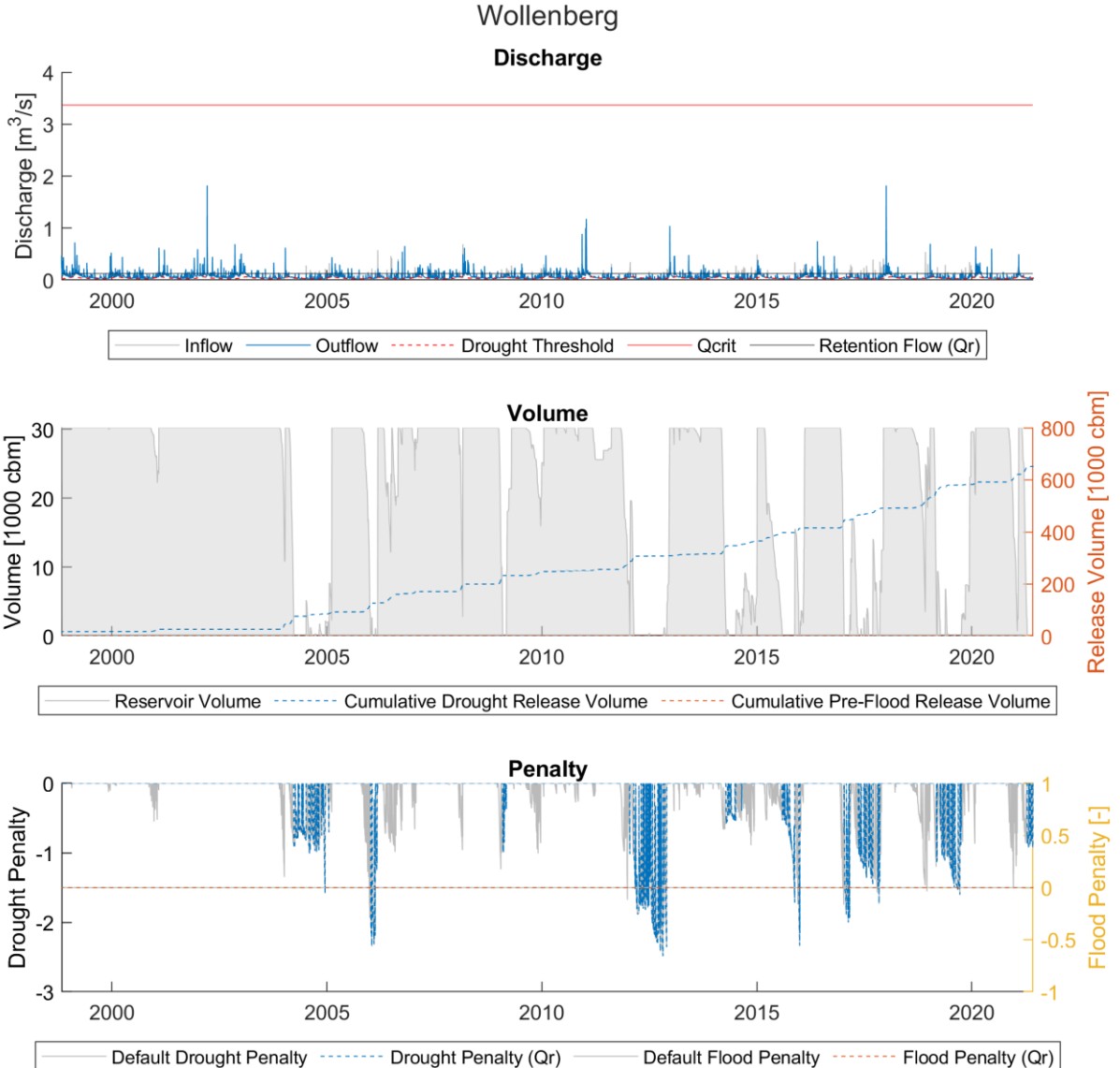

**Figure 10. Discharge, volume, and penalty time series for Wollenberg reservoir (example of a low availability, low benefit reservoir). The flooding limit ($Q_{crit}$ = 3.37 m$^3$ s$^{-1}$) is omitted in the discharge portion of this figure for clarity, as the flows never exceed 0.3 m$^3$ s$^{-1}$ during the 24 years of modelled data.**


## 3.2    Reservoir Results

### 3.2.1 Flood Protection

We reaffirm the maintenance of flood protection by tracking the total amount of time in floods, the total volume of all flood waves, and the flood penalty for the inflow, the flood-only model, and the optimized combined operation model (Figure 11). 10 reservoirs were able to retain all flood events—both volume and time—in the simulation period during the flood operation model. 11 reservoirs did not experience any flood events in the same period. These reservoirs maintained the same level of flood protection in the combined operation model—that is, they experienced no floods under combined operation. While nine reservoirs did experience flood failures in the flood operation model, the degrees of failure did not increase after optimizing the combined operation model. Thus, we demonstrate that it is possible to reuse these reservoirs for drought protection without impacting their flood protection functions.

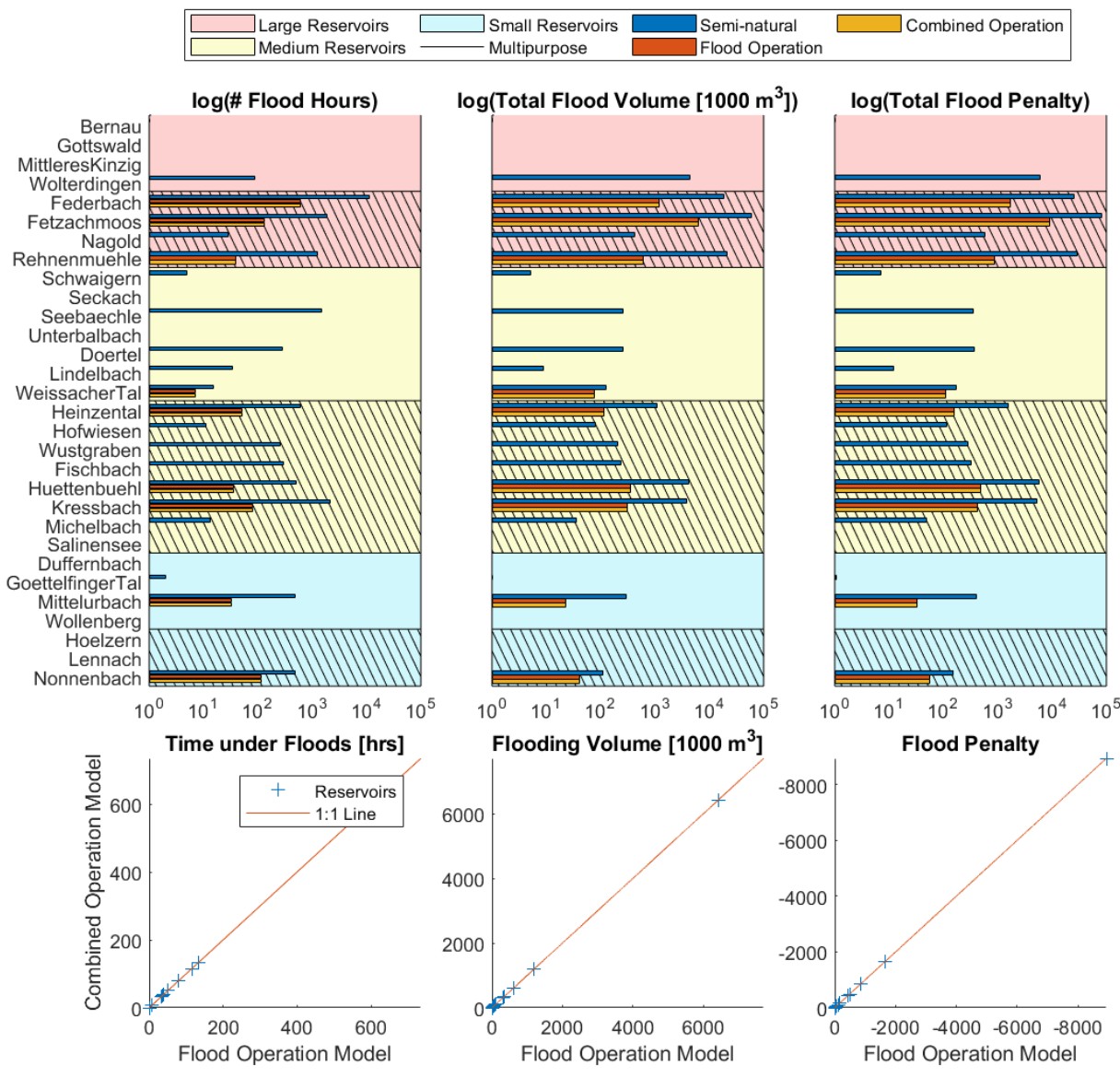

**Figure 11. Flood statistics (# of timesteps with floods, total flood volume, and flood penalty) for each of the 30 reservoirs at the inflow (semi-natural) and downstream under both models (flood operation and the optimized combined operation models). In the scatter plots (bottom), the x-values for each point denote the score in the flood operation model and the y-values denote the score in the combined operation model. A lower value in the combined operation model (i.e. deviation towards 0 from the 1:1 line) indicates improved performance. Note the differing axes and scales.**

### 3.2.2    Drought Protection

We plot similar metrics to evaluate the overall reduction of drought conditions in terms of hours, deficit volume, and penalty between different model runs (Figure 12). While we include the semi-natural condition for completeness, the focus in this discussion remains between the approximation of the current situation—the flood operation model—and the optimized combined operation model. Between the flood and combined operation models, there are significant reductions in time under drought for almost all reservoirs, while the reductions in deficit volume and penalty are not nearly as marked. This is again

due to the model releasing water from the reservoir as soon as the threshold is reached—because the deficit volumes at the beginning of a drought spell are smaller, the reservoir can supply water for longer. Changing the timing of releases to increase overall benefit would reduce the improvement in time. While this can be desirable, the purpose of the drought releases should also be considered: it may, for example, be more beneficial to alleviate drought conditions for longer if they happen to occur during critical times for agriculture or protected ecosystems. Interestingly, several reservoirs (Federbach, Lindelbach, and

Duffernbach) in the flood operation model result in an improvement in drought metrics compared to the inflow—in these cases, there were flood events that were immediately followed by drought conditions, so the immediate release of flood water happened to compensate for some drought deficits.

Drought penalty and drought deficit volume have a relationship that is significantly less straightforward than their flooding counterparts. For example, while the large flood-only reservoirs have the largest total deficits, they also have the smallest

penalties. This is because of the way that penalty adds "urgency" to the deficit volume: given equal deficit volumes, if the discharge is closer to zero, the (magnitude of the) penalty increases significantly. This adaptation is critical to ensuring that releases to flows that are low in both frequency (i.e. under the $Q_{70}$) and low in magnitude (i.e. low discharge) are properly valued. On the other hand, this means that if flows are high, the penalty for drought flows will not be high in magnitude. Thus, the penalty benefit is a clearer metric for analysis of the reservoir's performance.

The penalty and volume benefits are shown in Figure 14. The relationship between volume and penalty benefit here can be illustrative. Because of the "urgency" weighting, whether or not the penalty benefit is higher than the volume benefit may give an indication to how effective these release rules are. A higher volume benefit, for example, would imply that water was mostly given at less-critical times. This is the case for most of the reservoirs. The handful of reservoirs with relatively equal volume and penalty benefits may be able to satisfy critical deficits if the conditions are right, whereas the few with higher penalty

benefit can be considered quite effective in their release timings.

However, the reductions in deficit—in other words, the water the reservoir is able to supply—remain rather significant for most reservoirs (Figure 14). Flood pre-releases are also shown to contextualize how much water saved for drought is "lost" when maintaining flood protection. Multipurpose reservoirs have the highest pre-release volumes—this is likely due to their lower $Q_{crit}$, which is more frequently reached. Total drought release volumes range from 387 m$^3$ to 127 million m$^3$. The median

drought release volume is roughly 1.4 million m$^3$ over the simulation period, or approximately 58,000 m$^3$ per year. Assuming an average irrigation water demand (IWD) of 112 mm/year as found for crops in Germany by Drastig et al. (2016) (and also

assuming this water could be given at the right time), this median could fulfil the irrigation demand for half a square kilometre of farmland for 24 years. If all the reservoirs' drought releases were used purely for supplying this IWD, the water gained using the combined operation model could sustain almost 180 km$^2$ of agriculture per year. This has powerful implications for

satisfying agricultural demands in a warming world: these reservoirs could be used to provide needed irrigation.

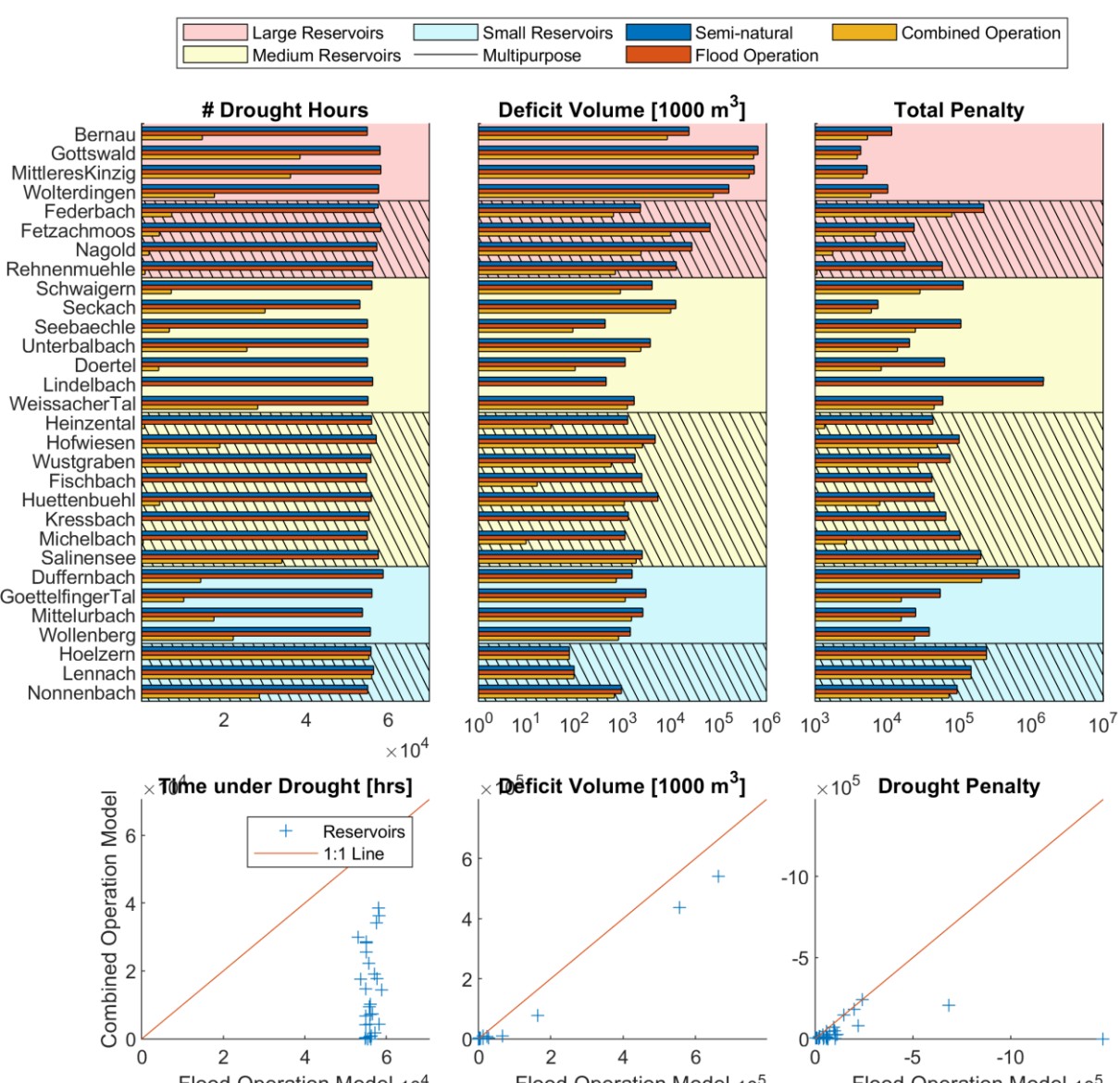

**Figure 12. Drought statistics (# drought timesteps, drought deficit volume, and drought penalty) for each of the 30 reservoirs at the inflow (semi-natural) and downstream under both models (flood operation and the optimized combined operation models). In the scatter plots (bottom), the x-values for each point denote the score in the flood operation model and the y-values denote the score in the combined operation model. A lower value in the combined operation model (i.e. deviation towards 0 from the 1:1 line) indicates improved performance. Note the differing axes and scales.**


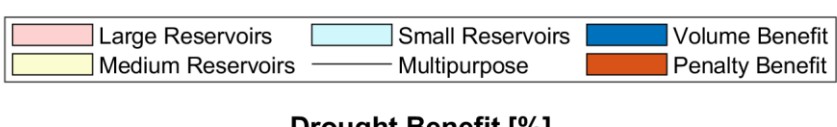

**Figure 13. Comparisons of volume and penalty benefit for all reservoirs.**

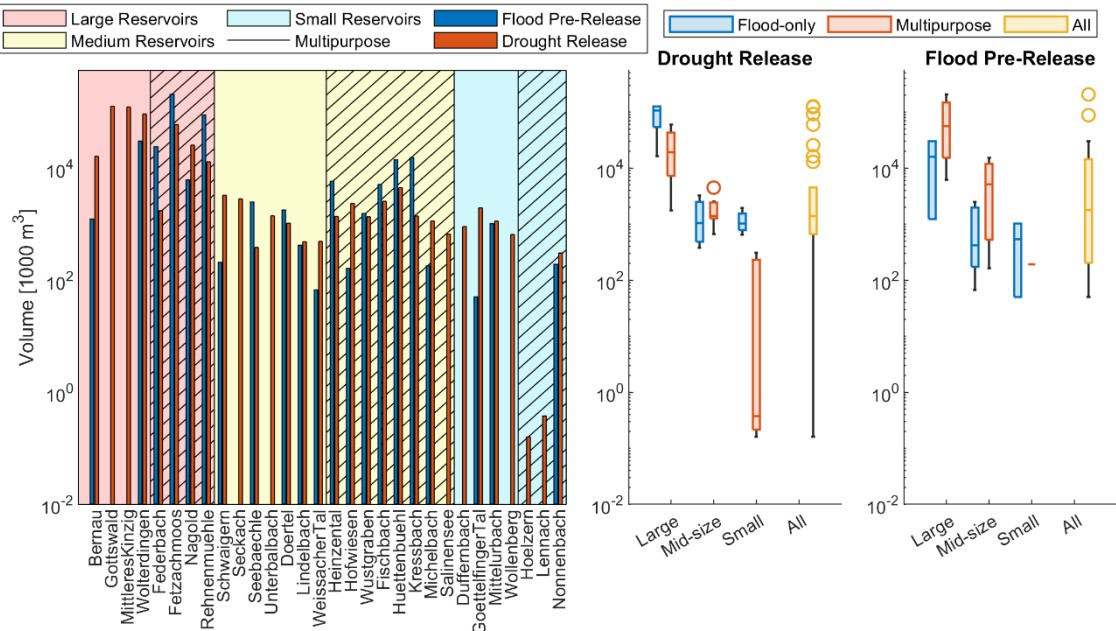

**Figure 14. Comparisons of total releases for drought protection and pre-releases for flood protection in the optimized combined operation model over the simulation period. Reservoirs with no flood pre-release volumes were omitted from the respective plots.**

### 3.2.3   SF and Reservoir Performance


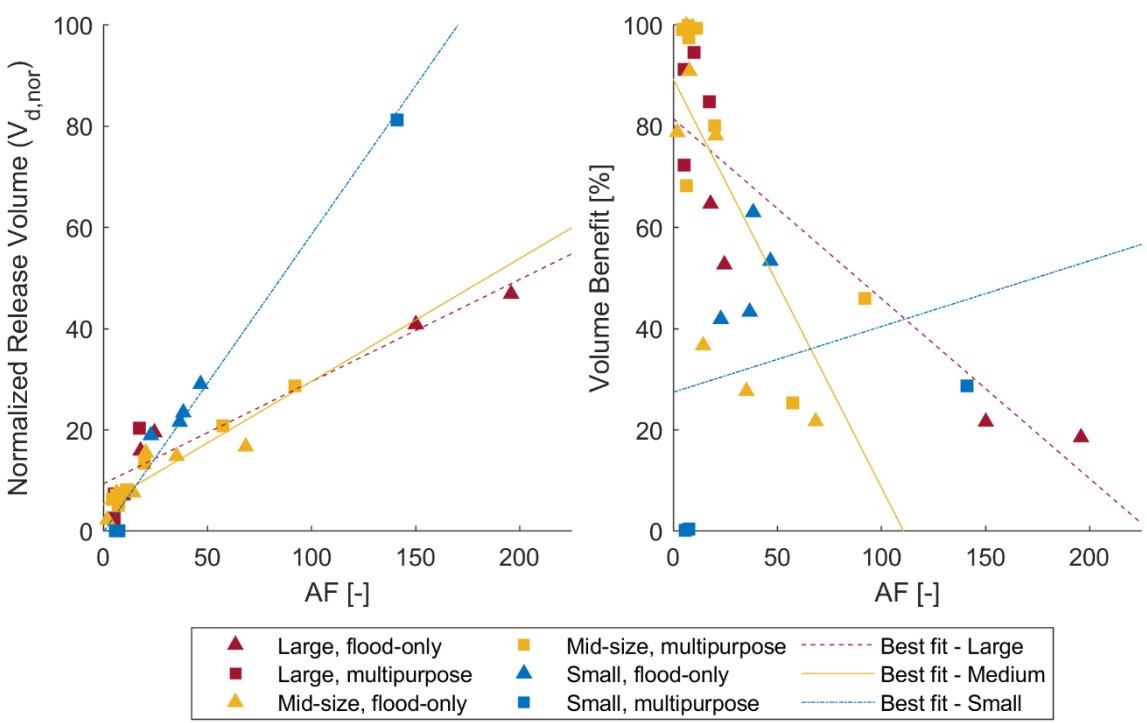

**Figure 15. The relation between AF (relative water availability) and normalized release volume ($V_{d,nor}$; left) and volume benefit ($B_v$; right).**

As we discussed in 3.1, reservoirs with a very high AF were overall unable to improve penalty benefit significantly (Figure 4,

left). This unfortunately remains the same for volume benefit (Figure 15, right), but seems to have a strong correlation with normalized release volume. In all reservoir categories, release volume increases with increasing availability. While AF has a relationship with release volume, having more water available and more water released does not correlate well with higher penalty benefit or deficit volume benefit.

In the large and medium size categories, benefit generally increases with decreasing AF. For example, the low improvement,

high release group consists exclusively of reservoirs with high AF. Even reservoirs with higher AF tend to have lower penalty reduction within their groups. A potential explanation is the chosen release rules: the model releases water as soon as inflows drop below the drought threshold. Because the deficits are small at first, the amount of penalty reduced per unit can be quite small. Changing the model so that the timing of reservoir releases such that water is given at the drought peaks could improve the penalty benefit further, though at the cost of complicating the model and the release rules. This would not, however,

improve the volume benefit. An alternative explanation for the disconnect between AF and benefit is a strong imbalance between incoming water and the capacity. This seems unlikely to improve, even if rules are significantly changed. For small reservoirs, the relationship between AF and benefit is skewed by the general unsuitability of SM reservoirs. The low penalty

benefit of these reservoirs is echoed in the low volume benefit. Overall, small reservoirs show a generally increasing volume benefit with increasing AF. This holds even if we do not consider the reservoirs with almost no benefit.

## 575  4    Conclusion

Under conditions of perfect knowledge, small (relative to typical reservoir studies) flood reservoirs can be repurposed for drought protection without impeding their flood protection functions. We expand the reservoir function by applying a retention flow above which we store water and supplying a drought threshold below which we release water, and maintain the flood functions by ensuring the reservoir is empty before a flood event. This method is a generalized framework through which flood
reservoirs—even those outside of our study area—can be evaluated for drought protection. We tested these new rules for a representative subset of 30 reservoirs to determine if water availability would be a suitable indicator for a reservoir's availability to mitigate streamflow drought conditions (defined in this study as the hourly 70th percentile exceedance). This hypothesis is built on the assumption that more available water for storage would mean that the reservoir would be more likely to have water available during drought conditions, and thus reduce drought more effectively. We found a range of results:
there are reservoirs that can release up to 80 times their capacity with limited benefit for streamflow drought prevention, others that can reduce streamflow drought conditions and water deficits by almost 95% over a 24-year simulation period, and still those that have potential but are limited by either the capacity or by constraints for flood protection. The median volume of water made available by this strategy across all reservoirs is approximately 1.4 million m$^3$, and the amount of streamflow drought reduction (benefit) ranges from nearly no effect to complete elimination of drought conditions.

Contrary to our hypothesis, the relative water availability (defined in this study as the availability factor, or the number of times per year that the reservoir can be filled using the difference between the mean and mean low flow) did not have a strong relationship to a reservoir's ability to curtail drought conditions. While it does have a strong relationship with the amount of water released for drought protection, the operation strategy of releasing water as soon as the drought threshold was reached meant that water was being delivered at less-than-optimal times. High relative water availability seems to indicate drought
conditions with considerable volume deficits for which the current reservoir volume cannot compensate, even if the retention flow were to be reduced further. Low relative water availability generally indicates milder drought conditions that can often be compensated by the reservoir's volume, resulting in high improvement. For mid-size and large reservoirs, the relationship between availability and benefit seems to be the inverse of what we expected—an exception are the small multipurpose reservoirs investigated in this study, which seem to function poorly under these operation rules. However, the overall lack of
generalizable rule indicates that water availability may not be a good predictor for drought reduction performance across all reservoir sizes.

Alongside the positive implications this work has for the role of repurposed flood reservoirs for increased water resources resilience, this work poses additional questions. For example, would the reservoirs still maintain their performance when operating under uncertain forecasts? This work, in considering a perfect-forecast scenario, provides a useful best-case scenario

of benefit which can serve as a benchmark for evaluating the impact of forecasting on reservoir operations. Reservoirs with frequent floods already see limitations on their drought reduction performance in these best-case scenarios. The potential benefits (and, perhaps, consequences) of operation with flood forecasts, which will inevitably have false alarms or misses, should also be investigated.

There remains additional questions regarding the true benefit and potential consequences of the provided water. We assumed
that any additional water volume—irrespective of nutrient quality or temperature—is beneficial. This is not necessarily the case, as fragile aquatic ecosystems could be damaged by an influx of poor quality water. Further work is needed to determine tangible benefits or even consequences of the water potentially supplied by these methods, as well as potential consequences from the land use change within the reservoir. Moreover, the benefit of the water in this study is based on streamflow statistics, with the assumption that any water provided during a streamflow shortage would be inherently beneficial, and that water
provided in dry-season shortages would be even more beneficial. While useful for demonstrating the potential of such a strategy under many different conditions of reservoir size and water availability (as was done in this study), this benefit unfortunately remains quite abstract and divorced from explicit consequences to the environment or human society. A clear next step in the development of a combined flood-drought strategy for small flood reservoirs is the investigation of the ability of combined operation to satisfy a particular demand, such as agriculture, in drought conditions.

# 5 Appendix

## 5.1 Appendix A

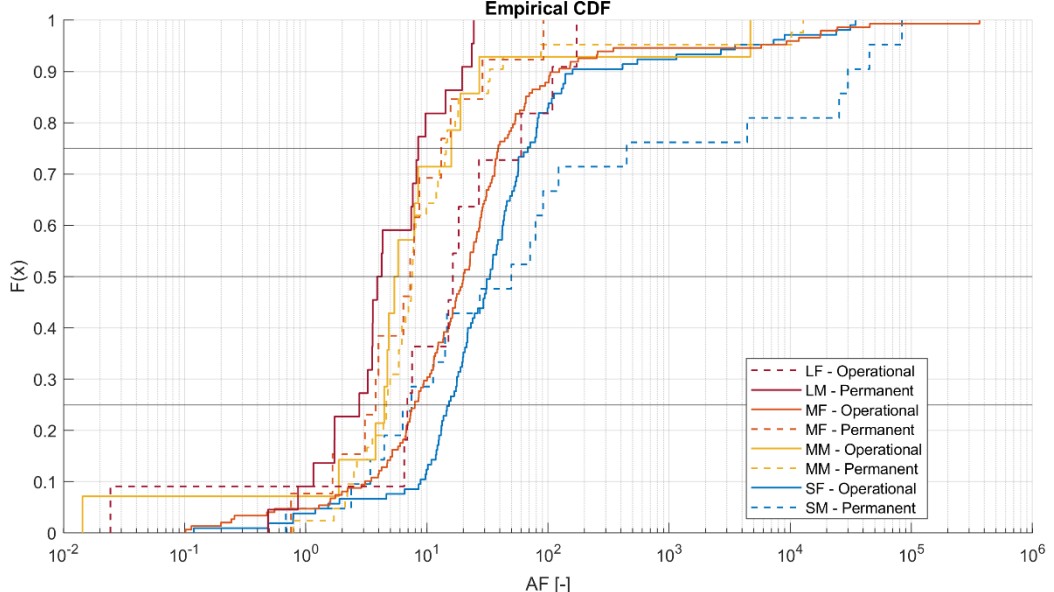


Appendix A1. Empirical CDF plots of AF for all reservoirs in the selected categories (see Table 1), estimated using local statistics. Reservoirs were selected based on their estimated AF, usually at the 25th, 50th, and 75th percentiles (and including the 87th and 12th percentiles if possible), to cover a broad range of water availability within a category.

## 5.2  Appendix B

Appendix B1. Flooding thresholds ($Q_{crit}$), maximum and minimum drought thresholds ($Q_{70}$), and optimal retention flow ($Q_r$) for each of the 30 reservoirs.

| Name | Category | Qcrit [m3/s] | Max(Q70) [m3/s] | Min(Q70) [m3/s] | Optimal Qr [m3/s] |
|---|---|---|---|---|---|
| Bernau | LF | 22.000 | 1.01 | 0.32 | 1.43 |
| Gottswald | LF | 75.000 | 20.62 | 4.93 | 36.81 |
| Mittleres Kinzigtal | LF | 830.000 | 16.99 | 3.81 | 33.85 |
| Wolterdingen | LF | 860.000 | 4.60 | 1.34 | 6.01 |
| Federbach | LM | 0.400 | 0.09 | 0.01 | 0.10 |
| Fetzachmoos | LM | 15.000 | 1.52 | 0.61 | 1.79 |
| Nagoldtalsperre | LM | 7.000 | 0.86 | 0.23 | 1.15 |
| Rehnenmuehle | LM | 15.000 | 0.52 | 0.05 | 0.65 |
| Schwaigern | MF | 0.790 | 0.13 | 0.02 | 0.20 |
| Seckach | MF | 0.500 | 0.75 | 0.26 | 1.74 |
| Seebaechle | MF | 2.410 | 0.01 | 0.01 | 0.02 |
| Unterbalbach | MF | 3.320 | 0.16 | 0.07 | 0.28 |
| Doertel | MF | 50.300 | 0.06 | 0.01 | 0.07 |
| Lindelbach | MF | 0.100 | 0.01 | 0.00 | 0.02 |
| Weissacher Tal | MF | 6.330 | 0.07 | 0.02 | 0.12 |
| Heinzental | MM | 3.700 | 0.06 | 0.02 | 0.08 |
| Hofwiesen | MM | 4.000 | 0.17 | 0.02 | 0.38 |
| Wustgraben | MM | 0.700 | 0.05 | 0.02 | 0.06 |
| Fischbach | MM | 1.000 | 0.10 | 0.03 | 0.17 |
| Huettenbuehl | MM | 3.600 | 0.23 | 0.04 | 0.30 |
| Kressbach | MM | 1.090 | 0.05 | 0.02 | 0.06 |
| Michelbach | MM | 10.680 | 0.04 | 0.01 | 0.06 |
| Salinensee | MM | 0.500 | 0.07 | 0.01 | 0.14 |
| Duffernbach | SF | 1.550 | 0.03 | 0.00 | 0.06 |
| Goettelfinger Tal | SF | 4.100 | 0.15 | 0.02 | 0.23 |
| Mittelurbach | SF | 0.500 | 0.09 | 0.06 | 0.10 |

| | | | | | |
|---|---|---|---|---|---|
| Wollenberg | SF | 3.370 | 0.06 | 0.02 | 0.13 |
| Hoelzern | SM | 1.500 | 0.00 | 0.00 | 0.03 |
| Lennach | SM | 2.100 | 0.00 | 0.00 | 0.05 |
| Nonnenbach | SM | 0.170 | 0.03 | 0.01 | 0.03 |

## 6 Code and Data Availability

The map in Figure 1 was created using ArcGIS® software by Esri with map data from OpenStreetMap (openstreetmap.org/copyright). ArcGIS® and ArcMap™ are the intellectual property of Esri and are used herein under license. Copyright © Esri. All rights reserved. All the relevant data for the reservoir models (semi-natural inflow results from LARSIM, $Q_{70}$ target time series, reservoir parameters, and outflow time series), as well as the developed code to run / optimize the reservoir models, are available through https://doi.org/10.5281/zenodo.12724797.

## 7 Author Contribution

Sarah Quynh-Giang Ho (SQH) and Uwe Ehret (UE) conceived and designed the methodology and reservoir models, which was coded, implemented, and executed by SQH. Data analysis was performed primarily by SQH, with input and guidance from UE. SQH wrote the initial draft of the paper. UE supervised the research and contributed to the improvement of the paper.

## 8 Competing Intersts

The authors declare that they have no conflict of interest.

## 9 Acknowledgements

We would like to thank the editor and the reviewers for the time, effort, and comments they contributed to this paper. This work was completed with data and invaluable support from Chantal Kipp and Olga Kiseleva from the Landesanstalt für Umwelt Baden-Württemberg, Dr. Hans Göppert and Johannes Höfer from WALD+CORBE Consulting GmbH, and Dr. Frank Seidel from IWU-WB at KIT. Sarah Ho would like to thank Prof. Jay Lund, Dr. Sarah Yarnell, and Dr. Francisco Bellido Leiva from the Center for Watershed Sciences at the University of California, Davis, as well as Prof. Kerstin Stahl and the colleagues from the Albert-Ludwigs-University of Freiburg for the invigorating discussions that sparked some of the ideas in this paper. Finally, we would like to thank the people of Baden-Württemberg who, through their taxes, provided the basis for this research.

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
