# Peer review of "Is drought protection possible without compromising flood protection? Estimating the potential dual-use benefit of small flood reservoirs in Southern Germany"

_EGUsphere, 2024_

## Author Response (AR1)

Dear Editor, dear Reviewer 1,

Please find our point-by-point reply to the reviewer comments below. For easier reading, the reviewer comments are in blue, our replies are in standard font, and the excerpts from the manuscript revisions are in *italics*.

The manuscript by Ho and Ehret investigates the potential of small, medium and large reservoirs designed for flood control (flood protection or multipurpose reservoirs) to mitigate drought. They select 30 examples from more than 600 existing reservoirs in Baden-Württenberg (Southwest Germany) and apply a "perfect runoff prediction" into these remaining reservoirs by using the conceptual hydrological model LARSIM. These inflows to the reservoirs are labelled "semi-natural". Based on this inflow over a 24 year daily time series, the authors tried different operating rules for flood retention and drought release. The prerequisite for this was that flood protection should not be impaired by the new operating rules. The operating rules are based on two-point hedging rules (where hedging storage begins at one point and ends at another).

The idea using flood retention basins to mitigate droughts is worth exploring. However, the manuscript has significant flaws. I would recommend rejecting the paper and resubmitting it after a complete revision.

We thank the reviewer for their time in writing these comments. However, we feel that the reviewer's rejection of the manuscript is based in a fundamental misunderstanding of our work. It seems that the reviewer is under the impression that our work aims to provide a comprehensive plan to mitigate drought in the study area using a network of flood retention basins. This is not the case. Rather, we seek to demonstrate that pre-existing individual flood basins can be repurposed for alleviation of drought conditions downstream. We have revised our manuscript to address this misunderstanding. We also would like to advise the reviewer that we have updated figures 4-14, in addition to adding one more, after re-running the models with some revisions suggested by another reviewer. The overall conclusions remain the same, though there are some details that have changed.

My main concerns are the following (see more details below):

1. The connection between drought mitigation and water release is solely based on Q70. This is too simple to draw conclusions about drought defense.

While we agree that drought is an inherently complex and multivariable phenomena that, in general, cannot be defined by a single observed variable, we would like to maintain that Q70 is a simple yet comprehensive method to define drought conditions. We have added the following discussion to the methods section:

Lines 241-257

*Drought remains a complex and multivariate phenomenon that affects multiple sectors, though different users may experience these effects at different times. This makes drought difficult to quantify and define. A distinction should be drawn between drought events and drought conditions: drought conditions are levels of intense dryness below a certain threshold, whereas drought events are prolonged periods of drought conditions (usually with a minimum duration of 30 days). Given that a reservoir's most immediate impact is on streamflow, we focus here on its potential ability to decrease streamflow drought conditions via streamflow drought thresholds as*

*a preliminary step into its ability to reduce drought. A truly comprehensive drought reduction approach would not just consider hydrologic variables but also consider management techniques (which is beyond the scope of this paper) for soil moisture, agricultural, and ecological drought within a given catchment to manage the prolonged dryness. However, hydrological droughts still have implications for impacts on other sectors, such as reduced drinking water or irrigation water availability (Van Loon, 2015), and many healthy ecosystems depend on certain flows at certain times (Yarnell et al., 2020). Streamflow drought, often expressed as a threshold level, is a common hydrological drought indicator.*

*Because such thresholds can be extremely variable and location-specific, especially for reservoir flows, a method that could be applied to different 30 reservoirs was needed. This method should also allow for seasonal variability, as previous studies on reservoir hedging rules for preventing drought have demonstrated that such rules are most effective when allowed to vary throughout the year (Chang et al., 1995; Balley, 1997). The drought release targets in this study should therefore be a streamflow drought threshold that allows for seasonal variability that could be applied anywhere.*

2. The spatial and hydrological context of the (arbitrary chosen) reservoirs is missing. Drought protection requires a model for river basin management.

Based on this and the previous comments, we are under the impression that there is a misunderstanding about the purpose of these reservoirs and of our work: this study is meant to assess, from a water supply perspective, how much drought mitigation benefit each existing (flood retention) reservoir working independently can produce. The goal is not to present a plan for complete water resources management in the area. We have clarified this in the following revisions:

Lines 102-103

*In this study, we seek to demonstrate the potential water supply benefit of converting pre-existing small (in the global context) flood retention basins into combined flood-drought reservoirs without impacting their flood protection functions.*

Lines 244-249

*Given that a reservoir's most immediate impact is on streamflow, we focus here on its potential ability to decrease streamflow drought conditions via streamflow drought thresholds as a preliminary step into its ability to reduce drought. A truly comprehensive drought reduction approach would not just consider hydrologic variables but also consider management techniques (which is beyond the scope of this paper) for soil moisture, agricultural, and ecological drought within a given catchment to manage the prolonged dryness.*

The main spatial and hydrological context of our work is the German state of Baden-Württemberg, which provides a common hydroclimatic background that is discussed in the paper. The hydrological context is expressed via the AF (formerly called SF, which we have adapted due to the reviewer's suggestion)—because the AF is based on the mean annual flow and the average low flow (i.e. the Q70), the AF aims to summarize the local hydrological context and relate it to the reservoir's capacity. This is noted in the following revisions:

Lines 161-162

*Reservoirs with different degrees of relative water availability from each of the categories were selected to investigate the various hydrological regimes within the region.*

Lines 167-168

*The AF can be interpreted as a combined indicator representing the relationship between the water availability in the catchment and the reservoir's ability to store or release it.*

We aimed to choose reservoirs with a range of physical (volume) and hydrological (summarized by the AF) characteristics that would represent the existing reservoirs in Baden-Württemberg, making the choice less arbitrary. We further clarify this in the following revisions:

Lines 167-176

*The AF can be interpreted as a combined indicator representing the relationship between the water availability in the catchment and the reservoir's ability to store or release it. A higher AF, then, indicates more water availability relative to the reservoir's capacity. While a reservoir's capacity inherently limits its ability to regulate streamflow, more available water should allow a reservoir to refill more quickly after emptying. In essence, it would increase the likelihood that, in drought conditions, a reservoir would have water to release. This assumption is the basis for our hypothesis that a reservoir with a higher AF should be able to reduce drought conditions more effectively. To test this, we selected reservoirs with varying values of AF, estimated from local long term statistics (Baden-Württemberg, 2016), from each category. For each combination of category and inundation type, reservoirs whose estimated AFs were close to the $50^{th}$, $25^{th}$, and $75^{th}$ percentile were selected (the distributions of estimated AF for each of the categories can be seen in Appendix 1). A selected few other reservoirs (Gottswald, Mittleres Kinzigtal, and Fetzachmoos) were selected based on stakeholder interest.*

3. The parameter SF is not helpful (as the authors admit). Why didn't they choose a different parameter?

The AF is a combined indicator representing the number of times the reservoir can be filled throughout a typical year, based on the difference between the average flow volume and the average low flow volume. This is a slightly modified version of a storage ratio (volume of inflow to capacity) used in the state of Baden-Württemberg to categorize flood retention basins. For this purpose, it has proven very useful, and therefore using a modified version of it for reservoir categorization for flood/drought was a natural first choice and a reasonable hypothesis. Our modified AF estimates how much water exists above the streamflow drought threshold in a typical year. When conceptualizing a study to determine relevant characteristics that would indicate suitability for storage of flood volume for drought usage, we hypothesized that a reservoir that had more available water to store—in other words, a higher AF—would have a greater impact on the downstream drought conditions. This is based on the assumption that more water available would also mean more water to release in drought conditions.

Testing this hypothesis also gave us another criteria to refine our reservoir selection without having to study all 600 reservoirs: by summarizing the hydrology of each reservoir by its inflow and normalizing it by the capacity, we gain an indicator that describes the hydrological conditions in a way that is comparable across different reservoir sizes. Our study showed,

however, that this was not entirely the case. While reservoirs with high AF were indeed able to release significant volumes of water during drought conditions, they were ultimately unable to significantly reduce total drought deficits (due to the total deficit being quite high) or drought penalty (due to shortages in summer being penalized more heavily than those in winter). Thus, while AF was ultimately not helpful as an indicator for potential drought penalty, it remains useful for characterizing water availability and potential water delivery relative to capacity. We have added the following discussions:

Lines 167-176

*The AF can be interpreted as a combined indicator representing the relationship between the water availability in the catchment and the reservoir's ability to store or release it. A higher AF, then, indicates more water availability relative to the reservoir's capacity. While a reservoir's capacity inherently limits its ability to regulate streamflow, more available water should allow a reservoir to refill more quickly after emptying. In essence, it would increase the likelihood that, in drought conditions, a reservoir would have water to release. This assumption is the basis for our hypothesis that a reservoir with a higher AF should be able to reduce drought conditions more effectively. To test this, we selected reservoirs with varying values of AF, estimated from local long term statistics (Baden-Württemberg, 2016), from each category. For each combination of category and inundation type, reservoirs whose estimated AFs were close to the 50th, 25th, and 75th percentile were selected (the distributions of estimated AF for each of the categories can be seen in Appendix 1). A selected few other reservoirs (Gottswald, Mittleres Kinzigtal, and Fetzachmoos) were selected based on stakeholder interest.*

Lines 592-596

*For mid-size and large reservoirs, the relationship between availability and benefit seems to be the inverse of what we expected—an exception are the small multipurpose reservoirs investigated in this study, which seem to function poorly under these operation rules. However, the overall lack of generalizable rule indicates that water availability may not be a good predictor for drought reduction performance across all reservoir sizes.*

    4. The conclusion "reservoirs can release up to 80 times their capacity and reduce drought penalties and water deficits by almost 95% over a 24-year simulation period" is mentioned twice (abstract and conclusions). That is the exception and not the rule and therefore misleading.

The statement about releases and penalty / deficit reductions is not meant to be a general conclusion valid for each reservoir, but rather a summary of the range of results. To avoid misunderstandings, we have revised the manuscript as follows:

Lines 30-33

*The optimized results were varied: there are reservoirs that can release up to 80 times their capacity with limited benefit for streamflow drought prevention; others that can reduce streamflow drought conditions and water deficits by almost 95% over a 24-year simulation period; and others that have potential but are limited by either the capacity or by constraints for flood protection.*

Lines 579-582

*We found a range of results: there are reservoirs that can release up to 80 times their capacity with limited benefit for streamflow drought prevention, others that can reduce streamflow drought conditions and water deficits by almost 95% over a 24-year simulation period, and still those that have potential but are limited by either the capacity or by constraints for flood protection.*

Further comments in detail:

1. Title: "…Estimating the maximum dual-use benefit of small flood reservoirs in Southern Germany" That is not correct. The authors also analyzed medium and large reservoirs.

We concede that the wording in our title can create a misunderstanding, as we do study reservoirs that are technically "large" in the sense that they are not strictly small reservoirs. However, even the "large" reservoirs here are, in comparison to the typical reservoir studied for this sort of usage, quite small—the largest reservoir in this study has a capacity of 4.3 million m^3, whereas typical reservoirs for this research have capacities that are at least an order of magnitude larger. In the interest of clarity, we have added the following to our introduction:

Lines 80-88

*More than 800 reservoirs in the German southwestern state of Baden-Württemberg exist today, with total capacities ranging from as small as 200 $m^3$ to almost 43 million $m^3$. 90% of these reservoirs have dams less than 15 meters in height. The German reservoir design standard DIN 19700 (Lubw, 2007) categorizes these reservoirs by dam height and capacity into large, medium, small, and very small reservoirs (see Table 1). In the global context, the majority of these "small" and "medium" would be small reservoirs. Many of the "large" reservoirs in Baden-Württemberg are just above the cutoff and remain quite small in comparison to typical large dams in the literature, which often have capacities that are at least an order of magnitude larger, generally 100 million to 1 billion $m^3$ (Consoli et al., 2007; Cañón et al., 2009; Liu et al., 2020). Henceforth we adopt the DIN 19700 size definitions as descriptors for reservoir sizes with the understanding that these refer to small and, at most, mid-size reservoirs on the global scale.*

2. Introduction: The sections on small reservoirs in Africa and worldwide do not fit the main content of the manuscript. Why do the authors write so much about agricultural (rainwater harvesting) reservoirs?

The section on small reservoirs worldwide serve to illustrate that reservoirs of that capacity are used for water resources. This is to show that there is precedent for this kind of work, as well as to introduce the challenge associated with reservoirs of this size. The references to agricultural reservoirs serve to demonstrate potential uses for the water and examples of existing reservoirs. We therefore keep this section in the manuscript.

To increase the logical connection between the small reservoirs worldwide and the discussion of reservoir sizes (see comment 1), we have moved the following from the study area section to the introduction:

Lines 94-101

*At the same time, drought events in Germany have been increasing in severity and frequency, including extreme events in 2018 and 2020 (Bundesamt, 2021; Erfurt et al., 2020). The potential shift in annual water availability in the near- and far-future due to both climate and*

*anthropogenic influences (Bundesamt, 2021) is the primary motivator for the state government's development of a 12-point plan for water shortages (Baden-Württemberg, 2021). The 12 actionable points fall under one of five categories: improving monitoring and information, managing and accounting of water uses, strengthening the resilience of existing water resources, improving awareness and protection incentives, and emergency planning. The potential reuse of flood reservoirs in this state for drought protection could contribute to improved resilience of water resources—provided, of course, that their flood retention capabilities are not impacted.*

3. The hypothesis "that the reservoirs providing the most benefit in drought conditions will be those that have high inflow relative to the reservoir capacity" is not justified. Why did the authors formulate this hypothesis?

Please also see our related reply to reviewer main comment 3. The hypothesis is built on the assumption that high inflow relative to reservoir capacity means that there is a lot of water that can be retained and released throughout the year. This is especially important in our reservoirs, as they must be completely empty before flood events in order to guarantee flood protection and since not every flood event will completely refill the reservoir. More water available implies that more water can be stored—in other words, potentially more water can be delivered in drought conditions, thus reducing streamflow drought. This has been re-emphasized in the following revisions:

Lines 113-117

*We hypothesize that the reservoirs providing the most relative benefit in drought conditions will be those that have high water availability relative to the reservoir capacity. While the limited capacity reduces the reservoir's overall potential benefit, more water available for storage means that a reservoir could potentially store and release its capacity multiple times in a year, increasing the likelihood that it will be able to provide water at critical times.*

Lines 169-172

*While a reservoir's capacity inherently limits its ability to regulate streamflow, more available water should allow a reservoir to refill more quickly after emptying. In essence, it would increase the likelihood that, in drought conditions, a reservoir would have water to release. This assumption is the basis for our hypothesis that a reservoir with a higher AF should be able to reduce drought conditions more effectively.*

4. Line 127 until 135: The "slot" definition is unclear. The final selection of the reservoirs is not explained.

The "slots" simply refer to number of allocated positions for the final selection. Selecting numbers of reservoirs based on proportion to the whole would result in either an overrepresentation of rather similar reservoirs or an unwieldy number of reservoirs; hence, we assigned each of the highly populated categories several "slots" or positions in the final selection. We clarify this in the revision:

Lines 152-153

*Each category containing 15 or more reservoirs was initially assigned three slots (i.e. a reservoir from this category will be chosen) for the reservoir selection.*

Lines 171-176

*This assumption is the basis for our hypothesis that a reservoir with a higher AF should be able to reduce drought conditions more effectively. To test this, we selected reservoirs with varying values of AF, estimated from local long term statistics (Baden-Württemberg, 2016), from each category. For each combination of category and inundation type, reservoirs whose estimated AFs were close to the 50[th], 25[th], and 75[th] percentile were selected (the distributions of estimated AF for each of the categories can be seen in Appendix 1). A selected few other reservoirs (Gottswald, Mittleres Kinzigtal, and Fetzachmoos) were selected based on stakeholder interest.*

5. The storage factor SF should be renamed as "availability factor AF". That would also avoid confusion with the variable designation SF already assigned for "small flood protection" (see Table 1). However, it is unclear why the storage factor can support drought mitigation. It is not logical that smaller C value (i.e. a higher SF value ) will "reduce drought conditions more effectively".

Thank you for this suggestion for a change of naming. We have adopted this in the manuscript.

The storage factor is the number of times per year that the reservoir can be filled to capacity—in other words, there are more chances to save water for drought conditions. We have given the relevant revisions in the responses to the reviewer's "main" comments 2 and 3 and in "detailed" comment 3.

6. Table 2: the volume of C should be added.

Thank you for the suggestion. We have moved the capacity from Table 3 to Table 2 and moved Table 3 to the appendix, as suggested in a later comment.

7. Equation 3: Why "-5" as a penalty factor?

The only constraint for the penalty factor is that it should be negative; the exact number is arbitrary for its functionality, as it is a simple linear transformation. This has been added:

Line 319-321:

*Here, it is a linear transformation (arbitrarily given a slope of 5) of flooding downstream of the river where penalty increases significantly once the outflow $Q_{out,t}$ exceeds the flooding discharge…*

8. Equation 3 and 4: Create a diagram for a better interpretation of the penalty functions.

We feel that the equations already provide all necessary information, but have added a brief discussion and example of the drought penalty:

Lines 327-331

*Because penalty at t is a function of the seasonally variable $Q_{70}$ at that time step, penalty also has an element of seasonality. The penalty per missing unit volume of water changes with $Q_{70}$: it will be higher in seasons where $Q_{70}$ (and therefore streamflow in general) is low, and lower in seasons where $Q_{70}$ is high. For example, the penalty of missing 1 $m^3/s$ if $Q_{70}$ is 2 $m^3/s$ is -0.293; if $Q_{70}$ is 10 $m^3/s$, the penalty is -0.0171. In this way, the model correctly penalizes shortages in the dry season more heavily than during the wet season.*

9. Table 3: Move it to the appendix.

Please see our response to comment 6.

10. Figure 4: Why is the penalty benefit of SMALL multipurpose reservoirs so much lower than flood-only reservoirs? Please explain and discuss!

Thank you for this question. We have added a more detailed discussion:

Lines 380-386

*SM reservoirs, however, are almost completely ineffective. In the cases of Hoelzern and Lennach, it is because the reservoirs had a $Q_{crit}$ that was so high in comparison to the modeled inflow—in the 24 years of simulation, they were generally on the scale of 0.005 $m^3/s$ and did not experience a single flood wave large enough to impound water. Even the lowest $Q_r$ value tested barely allowed for any storage of water in the combined scenario. Thus, there was almost no water available to release. Even when there was water available, as in the case of Nonnenbach, there was simply not enough volume in the reservoir to compensate for the drought deficits. Because of this, we consider these reservoirs generally unsuitable for a combined use strategy under these conditions.*

11. Figure 5 until 9: The flood penalty can be skipped because the precondition was that flood protection must not be impaired. Furthermore, the graphs related to outflow and drought penalties are not clearly recognizable.

We would like to maintain that the flood penalty carries relevant information for the reader about the complete interaction of the system (how flooding affects volume, which affects penalty), e.g. in Figure 8, and choose to keep these. We have also slightly modified the plots in hopes of improving readability.

12. Figure 10: Can be skipped (flood statistics are not changed due to the authors' precondition)

We would like to maintain that the flood penalty statistics provide context for high flood pre-releases and differences in performances between reservoirs.

13. Line 440 ff: The argumentation with the IWD is very weak. Either omit or elaborate in detail (seasonal influence, PET, soil conditions etc).

The discussion with the IWD is intended to contextualize how much water is made available through our strategy and what real world impacts it could have with a callback to the agricultural small reservoirs in the introduction. We think this has value for the reader and therefore prefer to

keep it in the manuscript, as it also introduces the potential for future work. We have briefly added to this link in the following revisions:

Lines 535-536

*This has powerful implications for satisfying agricultural demands in a warming world: these reservoirs could be used to provide needed irrigation.*

Lines 608-614

*… the benefit of the water in this study is based on streamflow statistics, with the assumption that any water provided during a streamflow shortage would be inherently beneficial, and that water provided in dry-season shortages would be even more beneficial. While useful for demonstrating the potential of such a strategy under many different conditions of reservoir size and water availability (as was done in this study), this benefit unfortunately remains quite abstract and divorced from explicit consequences to the environment or human society. A clear next step in the development of a combined flood-drought strategy for small flood reservoirs is the investigation of the ability of combined operation to satisfy a particular demand, such as agriculture, in drought conditions.*

14. Line 473: "Changing the model so that the timing of reservoir releases such that water is given at the drought peaks could improve the penalty benefit further, though at the cost of complicating the model and the release rules." This is recommended for the resubmission of the paper.

While we agree that this would be useful and interesting, we would like to argue that this is beyond the current scope of the paper. In this paper, we aim to a) determine whether or not the idea of reusing flood basins for drought protection is viable; and to b) determine whether or not water availability (via AF) is a suitable indicator for a reservoir's potential impact under these schemes. We do, however, plan to explore this potential change in releases—as well as more realistic demand targets tailored to local conditions—in a further paper.

Minor comments:

1. Line 253: "is used only used to"--> "is only used to"

Thank you; this has been corrected.

2. Equation 11 "V = C >=0" --> "V-C >=0"

We have revised the condition.

3. Line 330: "former" --> "latter"

This has been corrected.

4. Line 338: "flood droughts" --> "flood"

This has been corrected

5. Figure 11: Are the x- and y-axis swapped? "combined operation model" should be on the y-axis?

Thank you; this has been corrected.

6. Better to use Bp (Benefit for Drought) instead of Vd,nor (y-axis, left figure)

If this is in reference to Figure 14, we respectfully disagree—a plot of Bp against AF already exists in Figure 4. Moreover, this pair of figures serves to demonstrate that, while increasing water availability is strongly correlated with a higher Vd,nor, the total volume benefit remains variable among different reservoirs. We therefore will keep the figure as is.

Dear editor, dear reviewer 2,

We thank you for your time and thoughtful comments. Please find our point-by-point reply, accompanied by the most relevant revisions, to the reviewer's comments below. We also would like to inform the reviewer that, in accordance to another reviewer's comment, we have renamed the storage factor (SF) to the availability factor (AF) to avoid confusion with the small flood-only reservoir category (also SF).

For easier reading, the reviewer comments are in blue, our replies are in standard font, and the excerpts from the manuscript revisions are in *italics*.

General comments:

1. The drought definition of Q_70 as a static concept without consideration of dynamics, such as river flashiness, leads to the selection of unrealistic discharges, as the authors state in line 333f, on which the entire model optimization is based. A reconsideration of this definition, e.g. based on catchment type or other catchment characteristics, may be necessary.

We acknowledge that there are limitations associated with using only Q70 for drought, and that this definition is insufficient for a genuine application of our strategy. However, for the purposes of building up and testing the potential of this revised strategy, we felt that a preliminary definition based on statistics that could be calculated for many different locations would be a reasonable place to start, as local low-water warnings are based on a similar threshold. It has also been used as a threshold for water scarcity in previous studies (Van Loon et al., 2010; Hisdal et al., 2004; Van Loon and Van Lanen, 2012; Cammalleri et al., 2016) and to characterize flow regimes, including for ecological flows (Vigiak et al., 2018; Knight et al., 2011). We use this under the assumption that water levels below this threshold means there is some user of the water—whether human or otherwise—is suffering due to a reduced water level. We address the limitations of a streamflow threshold and further justify our choices in the following revisions:

Lines 241-257

*Drought remains a complex and multivariate phenomenon that affects multiple sectors, though different users may experience these effects at different times. This makes drought difficult to quantify and define. A distinction should be drawn between drought events and drought conditions: drought conditions are levels of intense dryness below a certain threshold, whereas drought events are prolonged periods of drought conditions (usually with a minimum duration of 30 days). Given that a reservoir's most immediate impact is on streamflow, we focus here on its potential ability to decrease streamflow drought conditions via streamflow drought thresholds as a preliminary step into its ability to reduce drought. A truly comprehensive drought reduction approach would not just consider hydrologic variables but also consider management techniques (which is beyond the scope of this paper) for soil moisture, agricultural, and ecological drought within a given catchment to manage the prolonged dryness. However, hydrological droughts still have implications for impacts on other sectors, such as reduced drinking water or irrigation water availability (Van Loon, 2015), and many healthy ecosystems depend on certain flows at certain times (Yarnell et al., 2020). Streamflow drought, often expressed as a threshold level, is a common hydrological drought indicator.*

*Because such thresholds can be extremely variable and location-specific, especially for reservoir flows, a method that could be applied to different 30 reservoirs was needed. This method should also allow for seasonal variability, as previous studies on reservoir hedging rules for preventing drought have demonstrated that such rules are most effective when allowed to vary throughout the year (Chang et al., 1995; Balley, 1997). The drought release targets in this study should therefore be a streamflow drought threshold that allows for seasonal variability that could be applied anywhere.*

Lines 270-275

*The choice of percentile has a notable effect on the detection of streamflow droughts: a higher percentile exceedance generally means more intense drought conditions and fewer drought events (Cammalleri et al., 2016; Tallaksen et al., 2009). We use the $70^{th}$ percentile exceedance flow ($Q_{70}$) as the threshold here, as it is the most lenient definition among typical values and allows insight on the reservoirs' ability to mitigate not only severe drought conditions but also mild ones. If the inflow at any time step is less than $Q_{70}$ (i.e. the discharge drops below the threshold), we assume that there is some user of the water—whether human or otherwise—that is being impacted by water scarcity in the river.*

More refined demand curves are currently under investigation as a next step in this work. In our revision, we have included this as follows:

Lines 605-614

*There remains additional questions regarding the true benefit and potential consequences of the provided water. We assumed that any additional water volume—irrespective of nutrient quality or temperature—is beneficial. This is not necessarily the case, as fragile aquatic ecosystems could be damaged by an influx of poor quality water. Further work is needed to determine tangible benefits or even consequences of the water potentially supplied by these methods, as well as potential consequences from the land use change within the reservoir. Moreover, the benefit of the water in this study is based on streamflow statistics, with the assumption that any water provided during a streamflow shortage would be inherently beneficial, and that water provided in dry-season shortages would be even more beneficial. While useful for demonstrating the potential of such a strategy under many different conditions of reservoir size and water availability (as was done in this study), this benefit unfortunately remains quite abstract and divorced from explicit consequences to the environment or human society. A clear next step in the development of a combined flood-drought strategy for small flood reservoirs is the investigation of the ability of combined operation to satisfy a particular demand, such as agriculture, in drought conditions.*

We also believe that the Q70 as applied in our study does allow for consideration of streamflow variability and flashiness. The Q70 is time-dynamic and is different for every hour of the year. By considering all discharges within a 30-day window centered on each time step, the resulting exceedance probability curves contain enough data as to provide a baseline for what is "normal" for that day and time. This would include river flashiness. We have addressed these in-text with the following revisions (lines 264-267):

*Typical reference values in the literature range from the 70-95$^{th}$ percentile (Hisdal et al., 2004; Cammalleri et al., 2016; Van Loon et al., 2010) and are generally adjusted for river dynamics (flashier rivers, for example, would typically select a higher percentile)—though in the interest of*

*consistent inter-reservoir comparisons, we will choose a singular percentile threshold for all reservoirs.*

However, we acknowledge that if the river is extremely flashy (as in the context of line 333), the exceedance probability curve will shift and a 70th percentile exceedance may not be strict enough. While for a redesign of such a reservoir it might be necessary to select a stricter definition (e.g. Q90 a.k.a. 90th percentile exceedance), we continue the use of the Q70 in order to compare the different reservoirs using the same low flow definitions. More specifically, we address our choice of keeping the Q70 despite the flashiness of Gottswald (the source of the unrealistic discharges discussion) in lines 414-418:

*At the same time, the large deficits are (at least in part) a result of the $Q_{70}$ as the drought definition: in a highly variable flow regime, this lenient definition may select flows that are unrealistically high for dry conditions. In these cases, it may be more realistic to choose a higher percentile exceedance for a more optimized operation. However, we retain the use of $Q_{70}$ so that the operation of different reservoirs in this study are analyzed at the same relative thresholds.*

2. The definition of SF is associated with some misleading assumptions. The hypothesis that reservoirs with a high SF are less efficient does not seem surprising, as small reservoirs with a large catchment area (leading to a high SF) are unlikely to have a significant impact on water supply.

We agree that reservoirs that are small for their catchment area are rather unlikely to have significant impact on overall water supply, but we assert that it is worth seeing if they are able to have an impact on low flow supplementation directly downstream due also, for example, to timing. This is also the motivation behind considering penalty and volume benefit separately (for more, please see the answer to 12). The SF is modified such that it describes the typical amount of water above the low flow condition relative to capacity—in other words, it is a description of how much water we can store without causing water shortages. We feel it is reasonable to assume that reservoirs that can be refilled more often will be able to give more water in critical times. We further clarify the reasoning behind our assumptions in the following revisions:

Lines 113-117

*We hypothesize that the reservoirs providing the most relative benefit in drought conditions will be those that have high water availability relative to the reservoir capacity. While the limited capacity reduces the reservoir's overall potential benefit, more water available for storage means that a reservoir could potentially store and release its capacity multiple times in a year, increasing the likelihood that it will be able to provide water at critical times.*

Lines 169-172

*While a reservoir's capacity inherently limits its ability to regulate streamflow, more available water should allow a reservoir to refill more quickly after emptying. In essence, it would increase the likelihood that, in drought conditions, a reservoir would have water to release. This assumption is the basis for our hypothesis that a reservoir with a higher AF should be able to reduce drought conditions more effectively.*

Additionally, testing this hypothesis also gave us a criteria we could use to refine our reservoir selection without having to study all 600 reservoirs: by summarizing the hydrology of each reservoir by its inflow and normalizing it by the capacity, we gain an indicator that describes the hydrological conditions in a way that is comparable across different reservoir sizes. Selecting reservoirs of varying SF allowed us to test the potential of this strategy using cases that span a broad range of flow conditions and reservoir sizes. We have clarified this in the following revisions:

Lines 172-175

*To test this, we selected reservoirs with varying values of AF, estimated from local long term statistics (Baden-Württemberg, 2016), from each category. For each combination of category and inundation type, reservoirs whose estimated AFs were close to the 50$^{th}$, 25$^{th}$, and 75$^{th}$ percentile were selected (the distributions of estimated AF for each of the categories can be seen in Appendix 1).*

3. Since a perfect knowledge scenario for the future is not realistic, it would be interesting to define a time window of X days that the authors would need for optimal operating decisions. How tenable are the operating proposals in climate change? If there is no flooding in the simulation period, the following interpretation does not have any meaning.

While a perfect knowledge scenario is indeed not realistic, it is useful in that it gives us upper bounds of benefit based on past scenarios. We emphasize that this is a potential study, and having a benchmark best-case scenario (presented in this work) is useful for our next studies.

We also agree that defining a time window for optimal decision-making would be an interesting question. We plan to address this in another phase of this work, in which we use (historical) flood forecasts at these locations to explore the effect of uncertain flood operation on the potential benefits to the reservoirs. This next phase will simulate a more realistic operation, including the effects of false alarms introducing more potential flooding events. The results of this study will be used as a benchmark for comparison, and give useful insights on the value of good flood predictions for this strategy.

We have addressed this in the following revisions:

Lines 597-603

*Alongside the positive implications this work has for the role of repurposed flood reservoirs for increased water resources resilience, this work poses additional questions. For example, would the reservoirs still maintain their performance when operating under uncertain forecasts? This work, in considering a perfect-forecast scenario, provides a useful best-case scenario of benefit which can serve as a benchmark for evaluating the impact of forecasting on reservoir operations. Reservoirs with frequent floods already see limitations on their drought reduction performance in these best-case scenarios. The potential benefits (and, perhaps, consequences) of operation with flood forecasts, which will inevitably have false alarms or misses, should also be investigated.*

4. The selection of 30 of the more than 800 retention basins seems to be biased. It would be of great benefit to define a workflow to identify all retention basins with high penalty benefit (perhaps for future work).

We agree with the importance of defining a workflow to determine penalty benefit—indeed, we had initially hypothesized that SF would be a way to identify high-benefit reservoirs, hence its emphasis in this paper. Unfortunately, this did not work. However, in order to develop such a workflow, we needed to test a broad range of possible conditions.

We concede that there is an element of bias to our selection, as large reservoirs are statistically overrepresented. This is due to the assumption that larger reservoirs would be able to impact water supply more effectively, as well as due to increased interest from relevant stakeholders after introducing our ideas. However, outside of selection of categories and numbers of reservoirs, our selection was based on estimates of SF from available statistics from the local environmental agency with the aim of selecting a wide range of water regimes in comparison to reservoir sizes (approximated by SF). This allows us to test the potential of our strategy in a broad range of conditions. We have added this information in the following revisions:

Lines 173-176

*To test this, we selected reservoirs with varying values of AF, estimated from local long term statistics (Baden-Württemberg, 2016), from each category. For each combination of category and inundation type, reservoirs whose estimated AFs were close to the 50$^{th}$, 25$^{th}$, and 75$^{th}$ percentile were selected (the distributions of estimated AF for each of the categories can be seen in Appendix 1).*

5. I understand that the ecosystem functions of the river downstream of the reservoir are out of scope. However, the impact on the ecosystem in the reservoir, where e.g. the current land use is severely affected due to the prolonged inundation, should at least be mentioned.

We agree, though we stress that this is beyond the scope of this potential study. This has been addressed in the following revisions:

Lines 145-148

*…in addition to its potential implications for technical modifications, reservoirs with operational inundation are more likely to have additional complications related to the current land use (e.g. loss of arable land, impacts to reservoir ecosystems). However, because these concerns are not relevant for optimizing water supply, this characteristic is not used in this study but is included for completeness.*

Lines 604-612

*There remains additional questions regarding the true benefit and potential consequences of the provided water. We assumed that any additional water volume—irrespective of nutrient quality or temperature—is beneficial. This is not necessarily the case, as fragile aquatic ecosystems could be damaged by an influx of poor quality water. Further work is needed to determine tangible benefits or even consequences of the water potentially supplied by these methods, as well as potential consequences from the land use change within the reservoir. Moreover, the benefit of the water in this study is based on streamflow statistics, with the assumption that any water provided during a streamflow shortage would be inherently beneficial, and that water provided in dry-season shortages would be even more beneficial. While useful for demonstrating the potential of such a strategy under many different conditions of reservoir size*

*and water availability (as was done in this study), this benefit unfortunately remains quite abstract and divorced from explicit consequences to the environment or human society.*

6.  The limitation to 30 reservoirs could at least for some allow a pictorial representation with some additional features such as morphology.

We have reviewed the manuscript and found that this distracts from the point of the paper (investigation of the potential of a dual-use operation), as the reservoir's morphology is largely irrelevant to its operation.

7.  To avoid distortions at the beginning of the simulation period, the first year should be taken as a lead time, for example, or the starting condition of the empty basin should be redefined.

We agree that these possible distortions should be considered and have incorporated this warm-up period in the model for this revision. We have documented this in the following lines:

Lines 332-334

*For discussion of results between reservoirs, we evaluate the reduction of penalty and drought deficit volume between the combined operation model and the flood operation model. These comparisons are done without results from the first year of operation to allow for a warm-up period.*

8.  Model uncertainties and statistical evaluations of trends are missing.

We are uncertain which statistical evaluations of trends are desired here, but in the case that streamflow trends are meant, we feel this is not entirely relevant, as trend analysis when calculating Q70 is not typically done. If statistics regarding the best-fit lines in Figure 14 were meant, we have reviewed the manuscript and feel that it is not particularly important to the key message of the paper.

We are also uncertain what model certainties are desired. For LARSIM uncertainties, please refer to the answer to 10; for uncertainties related to our assumption of perfect knowledge, please refer to the answer to 3.

9.  In the introduction, the context of the effects of drought in rivers is missing.

We agree that the context of drought effects is missing, but found that it fit better in the methods section as an introduction into our choice of streamflow drought threshold:

Lines 241-257

*Drought remains a complex and multivariate phenomenon that affects multiple sectors, though different users may experience these effects at different times. This makes drought difficult to quantify and define. A distinction should be drawn between drought events and drought conditions: drought conditions are levels of intense dryness below a certain threshold, whereas drought events are prolonged periods of drought conditions (usually with a minimum duration of 30 days). Given that a reservoir's most immediate impact is on streamflow, we focus here on its potential ability to decrease streamflow drought conditions via streamflow drought thresholds as*

*a preliminary step into its ability to reduce drought. A truly comprehensive drought reduction approach would not just consider hydrologic variables but also consider management techniques (which is beyond the scope of this paper) for soil moisture, agricultural, and ecological drought within a given catchment to manage the prolonged dryness. However, hydrological droughts still have implications for impacts on other sectors, such as reduced drinking water or irrigation water availability (Van Loon, 2015), and many healthy ecosystems depend on certain flows at certain times (Yarnell et al., 2020). Streamflow drought, often expressed as a threshold level, is a common hydrological drought indicator.*

*Because such thresholds can be extremely variable and location-specific, especially for reservoir flows, a method that could be applied to different 30 reservoirs was needed. This method should also allow for seasonal variability, as previous studies on reservoir hedging rules for preventing drought have demonstrated that such rules are most effective when allowed to vary throughout the year (Chang et al., 1995; Balley, 1997). The drought release targets in this study should therefore be a streamflow drought threshold that allows for seasonal variability that could be applied anywhere.*

    10. The authors state that LARSIM is typically not used for small catchment sizes. Could the authors explain the uncertainty arising from this and how they deal with it?

While LARSIM is not typically used for small catchment sizes, the pre-calibrated version that we are using is distributed on a 1 km$^2$ grid and therefore is, in principle, capable of doing small catchments. We further clarify that it is not used for operational forecasts—its main purpose—for catchments less than ~100 km$^2$ because of high associated uncertainties from weather forecasts, local conditions, etc.

We thank the reviewer, however, for this comment—after looking into the catchment sizes to ensure they aligned, we noticed a few errors in our model setup. We re-mapped the reservoirs to the relevant LARSIM subcatchments, seeking to match the catchment areas as closely as possible, and reran the hydrological model. However, due to different catchment delineation processes related to the 1 km$^2$ grid, a difference of a few km$^2$ can sometimes be unavoidable. We have adjusted for these differences by scaling the model output by the area ratio of delineated catchment area to the LARSIM catchment area.

These changes have been reflected in the following revisions:

Lines 193-203

*The model uses a grid structure with a 1 km$^2$ resolution to describe meso-scale hydrological processes such as interception, evaporation using the Penman-Monteith method, snow-related processes (accumulation, compaction, and melt), river routing, and soil water storage to evaluate discharge and water temperature. Thus, while typically used for large catchments (and calibrated to higher-order river discharges), it is also capable of modelling smaller headwater catchments by selecting the proper model output location. These model output locations were selected to have LARSIM-delineated catchments that are as similar as possible to actual conditions (e.g. connecting tributaries, catchment area). However, due to the 1 km$^2$ grid and different channel routing procedures, the LARSIM catchment area may differ from the true catchment area. We adjust for this by multiplying the resulting discharge by the ratio of true catchment area to LARSIM catchment area (the exception here being Fetzachmoos, whose*

*main structure as a diversion dam is not in the river network and whose delineated catchment area does not model the water that should be impounded).*

And as an addition to Table 2 via an additional column detailing the area scaling ratio.

Minor comments:

1. A lot of reading disruption due to hyphens.

We have attempted to amend this where possible.

2. Line 33ff: Sentence hard to understand.

We have revised this sentence for clarity:

*Research on optimal reservoir operation rules for drought have often focused on the concept of hedging rules. Hedging rules assume that by storing water and creating a small deficit of water now, we can use that water mitigate the consequences of a heavy deficit later (Shih and Revelle, 1994).*

3. Table 2: Horizontal lines between size categories would be needed.

This has been added.

4. Line 141f: Q_in and Q_70 are not flow volumes but volume rates.

We have clarified this in the revision.

5. Line 153f.: Large Area **Runoff** Simulation Model.

We have corrected this typo.

6. Sentence line 192 is for discussion not methods

We have reviewed the sentence and feel it is best placed as is.

7. Figure 2: Please update the figure without red underlining.

We have updated the figure.

8. Chapter 2.2.1 not quite logically structured (sentence line 216 could be after the introducing sentences in line 201 etc.).

We have restructured the chapter with all the comments in mind. In the interest of space, we have excluded the chapter—to see the revisions, please refer to the submitted manuscript.

9. Line 153: double used

We were unable to find the issue.

**10. Equation 3: Where does 5 come from?**

In this study, the exact factor in the flood penalty is arbitrary for its functionality, as it is only relevant for comparison to the flood-only operation. We have added it as follows:

Line 319-321:

*Here, it is a linear transformation (arbitrarily given a slope of 5) of flooding downstream of the river where penalty increases significantly once the outflow $Q_{out,t}$ exceeds the flooding discharge…*

**11. Line 233f: How does this fit in with integrated flood management?**

The flood protection offered by the reservoir is the total empty volume—we cannot do any better (as far as flood protection is concerned) than that. In order for us to maintain the planned flood protection of the reservoir, we must guarantee that the reservoir is empty before the event. We have addressed this in the following lines:

Lines 297-300

*After calculating $t_{down}$, the model checks if a flood begins ($Q_{in} > Q_{crit}$) within the next $t_{down}$ timesteps. This is, in effect, a perfect-knowledge flood forecast. If there is a flood, the model enters the pre-flood drawdown module in which the reservoir is emptied by releasing the water at $Q_{crit}$. Once emptied, the reservoir remains empty until the flood event begins. By ensuring that the flood reservoir is empty before onset, we guarantee that the flood protection function is not compromised.*

**12. Line 267f: How is weighting integrated in B_p?**

Thank you for the question; we have added some brief discussions in the following revisions:

Lines 327-331

*Because penalty at t is a function of the seasonally variable $Q_{70}$ at that time step, penalty also has an element of seasonality. The penalty per missing unit volume of water changes with $Q_{70}$: it will be higher in seasons where $Q_{70}$ (and therefore streamflow in general) is low, and lower in seasons where $Q_{70}$ is high. For example, the penalty of missing 1 $m^3/s$ if $Q_{70}$ is 2 $m^3/s$ is -0.293; if $Q_{70}$ is 10 $m^3/s$, the penalty is -0.0171. In this way, the model correctly penalizes shortages in the dry season more heavily than during the wet season.*

Lines 337-338

*Because penalty has an element of seasonality, the benefit per unit volume of water is also seasonal: the benefit associated with providing 1 $m^3/s$ will be higher when $Q_{70}$ is low than when $Q_{70}$ is high.*

As an aside, we would like to note that the equations for Bp and Bv have a typo. Bp should read:

$$B_p = 100 \times \frac{\sum P_{d,f} - \sum P_{d,c}}{\sum P_{d,f}}$$

And Bv has been changed accordingly.

> 13. Table 3: Decimal places could be reduced.

We have moved most of the content in Table 3 to the Appendix and reduced the decimal places.

> 14. Line 338: Flood droughts?

This is a typo and now reads "flood events"

> 15. Line 432: Figure reference doubled.

This has been corrected.

> 16. Line 475: The trend that the benefit increases with increasing SF only applies to flood-only basins (Figure 14).

We respectfully disagree. Though not as dramatic as flood-only basins, small multipurpose reservoirs do see increasing benefit with increasing SF.

> 17. Line 476ff: Replicate of line 362ff

Thank you; we have revised accordingly.

[revised manuscript text omitted]

---

## Referee Report (RR1)

**Is drought protection possible without compromising flood protection? Estimating the potential dual-use benefit of small flood reservoirs in Southern Germany**

The manuscript has been improved by emphasizing potential rather than optimal operation, and the value of the manuscript as a best-case scenario benchmark can be seen.

Minor comments (line numbering according to track changes manuscript):

- Please remain consistent with the spelling of south-western
- Line 129: Only small retention reservoirs (if the authors mean the global context; otherwise, the authors also look at large ones)
- Line 188: "Thus, we focus only on large, medium, or small reservoirs ..." is duplicative of your statement that you exclude very small reservoirs because then there are no other reservoirs left; and "thus" does not fit to the previous sentence.
- Line 210: If you calculate an AF as a medium for the entire simulation period (it would be helpful to specify the years here), please make this clearer.
- Line 211: rate rates
- Line 222f: double selected
- Table 1 (and general): Explanation of abbreviations LF, LM, MF ... is missing.
- Line 515: The sentence is not well structured.
- Figure 11: superscript m$^3$
- Figure 12: I think the axes in the scatter plots are switched. The first plot would otherwise suggest that for the combined operation model the time under droughts is always around 6x10^4, and for flood operation model ranging from 0 to 6x10^4.

---

## Author Response (AR2)

Dear editor,

Thank you for your efforts. We have edited the manuscript to correct the minor comments from reviewer 2 (their comments *in italics*, our response in normal font; line numbers and text referring to the tracked changes version). In addition, we would like to note that we added a minor change (the addition of the word "Runoff" in "Large Area Runoff Simulation model" in line 191) that was missing from the last revision and a few sentences to the acknowledgements.

> *Minor comments (line numbering according to track changes manuscript):*
> *- Please remain consistent with the spelling of south-western*

We have decided to use "southwestern" rather than "south-western". The manuscript should now be consistent.

Line 16:

... southwestern Germany alone...

> *- Line 129: Only small retention reservoirs (if the authors mean the global context; otherwise, the authors also look at large ones)*

We have re-clarified.

Lines 112-113:

... for a variety of DIN 19700 small, medium, and large flood retention reservoirs...

> *- Line 188: "Thus, we focus only on large, medium, or small reservoirs ..." is duplicative of your statement that you exclude very small reservoirs because then there are no other reservoirs left; and "thus" does not fit to the previous sentence.*

We have removed the sentence entirely, as it is indeed redundant, and made a small phrasing adjustment in the previous sentence.

Lines 142-145:

... leaving us with two purpose types: flood protection only, or multipurpose with flood protection, where flood protection-only reservoirs tend to have higher flooding thresholds than multipurpose ones. We also distinguish here...

> *- Line 210: If you calculate an AF as a medium for the entire simulation period (it would be helpful to specify the years here), please make this clearer.*

We have made small adjustments to properly denote this.

Lines 165-169:

We define relative water availability here as the availability factor (AF), or the average number of times per year that a reservoir's capacity (C, in cubic meters) can be filled via the water that we are able to store based on the entire simulation period (excluding the warm-up; i.e. 1998-2021). The water available for storage is the difference between the mean (calculated over the 24 years of

simulation) yearly inflow ($Q_{in}$) volume rate and the mean low flow ($Q_{70,mean}$; for definition and calculation see 2.4.1) volume rate, in cubic meters per second...

> - *Line 211: rate rates*

This has been deleted; see above.

> - *Line 222f: double selected*

We have revised the sentence.

Lines 178-179:

A few other reservoirs... were selected based on stakeholder interest.

> - *Table 1 (and general): Explanation of abbreviations LF, LM, MF ... is missing.*

We have added a brief explanation in-text and in the caption for Table 2.

Lines 158-159:

Categories are referred to in this study as a two-letter abbreviation combining their size (where L is large, M is medium, and S is small) and their usage (where F is a flood-only reservoir and M is a multipurpose reservoir).

Lines 162-163 (caption for Table 1):

Each category abbreviation is a combination of its size (large = L, medium = M, and small = S) and its usage (F = flood-only, M = multipurpose).

> - *Line 515: The sentence is not well structured.*

We have revised the sentence.

Lines 434-436:

Federbach (Figure 8) is a large multipurpose reservoir with a rather low AF in comparison to other large reservoirs. As it is on the lower end of the high-benefit reservoirs, it demonstrates some limitations that impact a reservoir's benefit.

> - *Figure 11: superscript m$^{3}$*

We have corrected this typo.

> - *Figure 12: I think the axes in the scatter plots are switched. The first plot would otherwise suggest that for the combined operation model the time under droughts is always around 6x10^4, and for flood operation model ranging from 0 to 6x10^4.*

The axes are not switched; each point is (Flood Operation Value, Combined Operation Value). We have added text offering interpretation assistance to the captions for figures 11 (as it is similar) and 12.

Lines 502-505:

In the scatter plots (bottom), the x-values for each point denote the score in the flood operation model and the y-values denote the score in the combined operation model. A lower value in the combined operation model (i.e. deviation towards 0 from the 1:1 line) indicates improved performance.  Note the differing axes and scales.

Lines 546-549 (identical to the above quote, but listed here for transparency):

In the scatter plots (bottom), the x-values for each point denote the score in the flood operation model and the y-values denote the score in the combined operation model. A lower value in the combined operation model (i.e. deviation towards 0 from the 1:1 line) indicates improved performance. Note the differing axes and scales.